The EMBO Journal (2013) 32, 2275–2286
www.embojournal.org

# Neuronal carbonic anhydrase VII provides GABAergic excitatory drive to exacerbate febrile seizures

Eva Ruusuvuori[1,5], Antje K Huebner[2,5], Ilya Kirilkin[1,5], Alexey Y Yukin[1], Peter Blaesse[1,6], Mohamed Helmy[1], Hyo Jung Kang[3,7], Malek El Muayed[2,8], J Christopher Hennings[2], Juha Voipio[1], Nenad Šestan[3], Christian A Hübner[2] and Kai Kaila[1,4,*]

[1]Department of Biosciences, University of Helsinki, Helsinki, Finland, [2]Institute of Human Genetics, University Hospital Jena, Friedrich-Schiller-University Jena, Kollegiengasse 10, Jena, Germany, [3]Department of Neurobiology and Kavli Institute for Neuroscience, Yale University School of Medicine, New Haven, CT, USA and [4]Neuroscience Center, University of Helsinki, Helsinki, Finland

**Brain carbonic anhydrases (CAs) are known to modulate neuronal signalling. Using a novel CA VII (*Car7*) knockout (KO) mouse as well as a CA II (*Car2*) KO and a CA II/VII double KO, we show that mature hippocampal pyramidal neurons are endowed with two cytosolic isoforms. CA VII is predominantly expressed by neurons starting around postnatal day 10 (P10). The ubiquitous isoform II is expressed in neurons at P20. Both isoforms enhance bicarbonate-driven GABAergic excitation during intense GABA$_A$-receptor activation. P13–14 CA VII KO mice show behavioural manifestations atypical of experimental febrile seizures (eFS) and a complete absence of electrographic seizures. A low dose of diazepam promotes eFS in P13–P14 rat pups, whereas seizures are blocked at higher concentrations that suppress breathing. Thus, the respiratory alkalosis-dependent eFS are exacerbated by GABAergic excitation. We found that CA VII mRNA is expressed in the human cerebral cortex before the age when febrile seizures (FS) occur in children. Our data indicate that CA VII is a key molecule in age-dependent neuronal pH regulation with consequent effects on generation of FS.**

*The EMBO Journal* (2013) **32**, 2275–2286. doi:10.1038/emboj.2013.160; Published online 23 July 2013
Subject Categories: neuroscience
Keywords: carbonic anhydrase expression; chloride accumulation; GABA$_A$ receptor; human brain; hyperthermia

*Corresponding author. Department of Biosciences, University of Helsinki, POBox 65, FI-00014 Helsinki, Finland. Tel.: +358 9 19159860; E-mail: Kai.Kaila@Helsinki.Fi
[5]ER, AKH and IK contributed equally to this work.
[6]Present address: Institute of Physiology I, Westfälische Wilhelms-University Münster, D-48149 Münster, Germany
[7]Present address: Department of Life Science, Chung-Ang University, Seoul, Korea
[8]Present address: Division of Endocrinology, Metabolism and Molecular Medicine, Northwestern University Feinberg School of Medicine, Chicago, IL 60611, USA

## Introduction

GABA$_A$-receptor (GABA$_A$R)-mediated signalling has a wide spectrum of functions in neurons and neuronal networks (Farrant and Kaila, 2007). Paradoxically, intense activation of GABAergic synapses in mature neurons may directly promote rather than suppress neuronal excitation (Alger and Nicoll, 1982; Kaila *et al*, 1997). We and others have previously demonstrated that, in the rat hippocampus, GABA$_A$R-mediated excitation is strictly dependent on the continuous replenishment of neuronal HCO$_3^-$ by cytosolic carbonic anhydrase (CA) activity and suppressed by membrane-permeant inhibitors of CA (Staley *et al*, 1995; Kaila *et al*, 1997; Fujiwara-Tsukamoto *et al*, 2007; Viitanen *et al*, 2010). Whether HCO$_3^-$-dependent GABAergic excitation has a role in the generation of seizures *in vivo* is not known.

CAs affect the kinetics and amplitudes of pH transients in distinct intra- and extracellular compartments (Chesler, 2003; Casey *et al*, 2010; Ruusuvuori and Kaila, 2013) and can thereby influence the function of a wide variety of proton-sensitive membrane proteins involved in neuronal signalling such as GABA$_A$Rs (Pasternack *et al*, 1996; Wilkins *et al*, 2005), *N*-methyl-D-aspartate receptors (Traynelis *et al*, 2010; Makani *et al*, 2012), acid-sensing ion channels (Waldmann *et al*, 1997), cation channels (Munsch and Pape, 1999; Williams *et al*, 2007; Enyedi and Czirjak, 2010) and gap junctions (Spray *et al*, 1981). In addition, pH transients can have modulatory actions on ion channels mediated by changes in fixed charges on the plasmalemmal surface (Velisek *et al*, 1994; Hille, 2001, see pp 653–654).

CA inhibitors such as acetazolamide and its more lipophilic derivatives have a long history as anticonvulsants, but the molecular targets and mechanisms of action at the neuronal network level are still poorly understood (Thiry *et al*, 2007; Supuran, 2008). The available inhibitors are not selective with respect to the 13 catalytically active isoforms that have been identified so far (Supuran, 2008) and may thus exert their actions within and outside the CNS. It is possible that their effects on neuronal functions are at least partly mediated by nitric oxide (Aamand *et al*, 2011) or calcium-activated potassium channels (Pickkers *et al*, 2001).

Using a novel CA VII knockout (KO) mouse as well as a CA II KO (Lewis *et al*, 1988) and a novel CA II/VII double KO, we show here that CA II and CA VII are the only cytosolic isoforms present in both somata and dendrites of mature hippocampal CA1 pyramidal neurons, with expression of CA VII commencing at postnatal day 10 (P10) and that of CA II at around P20. Immunoblots from glial or neuron/glial cultures showed that CA VII expression is restricted to neurons while CA II is present in both types of cells. After P30, the two isoforms are similarly effective in promoting GABAergic excitation.

In line with our previous data on rats (Schuchmann *et al*, 2006), hyperthermia generated a similar respiratory alkalosis in both wild-type (WT) and CA VII KO mice at P13–14. Typical experimental febrile seizures (eFS) with cortical electrographic seizure activity were observed in WT mice but not in the CA VII KOs. Diazepam potentiated excitatory GABAergic transmission *in vitro*, and a low concentration of the drug reduced the time to eFS onset in P14 rats *in vivo*. Consistent with a role in human febrile seizures (FS), we found that CA VII is present in the human cortex and hippocampus already at the perinatal stage, well before 6 months of age when FS are typically first observed (Berg and Shinnar, 1996; Stafstrom, 2002). Thus, CA VII is a key molecule in age-dependent neuronal pH regulation with consequent effects on generation of eFS.

## Results

### Generation of the CA VII KO mouse

We floxed exons 5–7 of the *Car7* gene (Figure 1A), and homologous recombination was verified by Southern blot analysis with a 3′ probe using an additional *EcoR*I site in the targeted allele. In two independent correctly targeted ES cell clones, we transiently expressed Cre-recombinase. The resulting subclones were screened for deletion of the DNA fragment flanked by the outer loxP sites. Two independent subclones were then injected into blastocysts and transferred to foster mice. The resulting chimeras were mated with C57BL/6 mice and produced offspring to heterozygous KO mice. Homozygous KO mice were born from heterozygous matings in the expected Mendelian ratio ($\sim 25\%$) and did not display any obvious phenotype within the age range used in the present experiments (up to P56, where P0 refers to the day of birth) or in the breeding animals used for about 1 year.

### Expression patterns of CA VII mRNA and protein

Northern blot analysis in adult mouse tissues showed that CA VII is prominently expressed in the mature brain (Figure 1B). To detect CA VII at the protein level, a novel polyclonal antiserum against an epitope of murine CA VII was raised in rabbits and affinity-purified. In protein lysates of adult WT whole hippocampi, this antibody detected a band of an appropriate size of $\sim 30\,kDa$, which was absent from protein lysates prepared from the hippocampi of CA VII KO mice (Figure 1C). In WT mice, CA VII protein abundance increased in the hippocampus during postnatal maturation. Immunoblots on CA VII from lysates of glial and mixed neuron/glia cultures showed that, unlike CA II, CA VII is expressed in neurons but not in glia (Figure 1D). Using an antibody that is not validated in CA VII KO tissue, Bootorabi *et al* (2010) reported expression of CA VII in numerous mouse tissues including skeletal muscle and liver, while no signal was seen in the CNS. These data are in stark contrast with the mRNA expression analysis shown in Figure 1B and also with immunoblots based on our antibody in muscle and liver where no CA VII protein was detected (Supplementary Figure S1).

### CA VII is the first functional cytosolic isoform expressed in somata of CA1 pyramidal neurons

Our previous results on rats showed a temporal coincidence between the developmental upregulation of CA VII mRNA and enhanced CA activity probed by CA inhibitors in pyramidal neurons (Ruusuvuori *et al*, 2004). Because the available cytosolic CA inhibitors are not isoform specific, these data do not exclude the possible presence of other neuronal CA isoforms. To overcome this problem, we examined the emergence of CA activity in somata of the hippocampal CA1 pyramidal neurons using BCECF fluorescence recording of intracellular pH ($pH_i$) in slices from neonatal, juvenile and adult WT and the three CA transgenic mice (CA VII KO, CA II KO and CA II/VII double KO) (Figure 2).

An abrupt developmental onset of cytosolic CA activity was detected in WT neurons at P10. The alkaline shifts measured in the somatic area upon $CO_2/HCO_3^-$ withdrawal became faster and larger and the membrane-permeant CA inhibitor acetazolamide caused a clear suppression of the maximum rate of the alkalinization in 81% of P10–18 neurons ($n = 53$) (Figure 2A and B). In sharp contrast to WT, functional cytosolic CA was not detected in CA VII KO neurons at this age. The alkalinization was slow and insensitive to acetazolamide in all of the $< P17$ CA VII KO neurons ($n = 60$) and in 28 of 29 P17–18 CA VII KO neurons, similar to the $pH_i$ response in WT neurons before the onset of CA expression ($n = 13$, WT neurons $< P10$) (Figure 2A and B). These results provide firm evidence for the conclusions that (i) the developmental upregulation of CA activity in the somatic area of mouse hippocampal CA1 pyramidal neurons at P10 is solely attributable to CA VII, and thus (ii) this is the *only cytosolic isoform* that shows catalytic activity in the P10–18 time window.

The expression of CA VII caused a small change in the baseline $pH_i$ (Supplementary Figure S2), which can be readily explained by production of $CO_2$ within the slice (Voipio and Kaila, 1993).

### Upregulation of cytosolic CA activity in somata of $> P18$ CA VII KO neurons is attributable to CA II

Surprisingly, an upregulation of CA activity was observed in neurons from CA VII KO mice at around P20 (Figure 2B and C). CA activity was observed in about 50% of $> P18$ neurons, and after P35 the percentage of CA VII KO cells showing catalytic activity was similar to that in WT.

Measurements of $pH_i$ in CA II/VII double KO neurons demonstrated that they were completely devoid of cytosolic CA activity even after P35. In contrast to this, CA activity was detected in a roughly similar fraction in WT, CA VII KO and CA II KO neurons after P35 (83, 88 and 80%). After P25, the baseline of somatic $pH_i$ was more acidotic compared with P14–15 neurons from WT and CA VII KO, and there was no longer a difference in the mean resting $pH_i$ between the two genotypes (Supplementary Figure S2), implying that the presence of both CA VII and CA II, or of CA II alone, had an identical effect on baseline pH. Thus, experiments with the three KO transgenic mice show that the cytosol of mature CA1 pyramidal neuron somata in WT mice is endowed with two CA isoforms, CA II and CA VII.

### Depolarizing dendritic GABA responses are promoted by CA II and CA VII

Next, we studied the developmental expression patterns of CA isoforms in the dendrites of pyramidal cells. To this end, we first used whole-cell recordings from CA1 pyramidal neurons in P12–16 WT and CA VII KO slices and examined GABAergic synaptic responses. Biphasic GABAergic

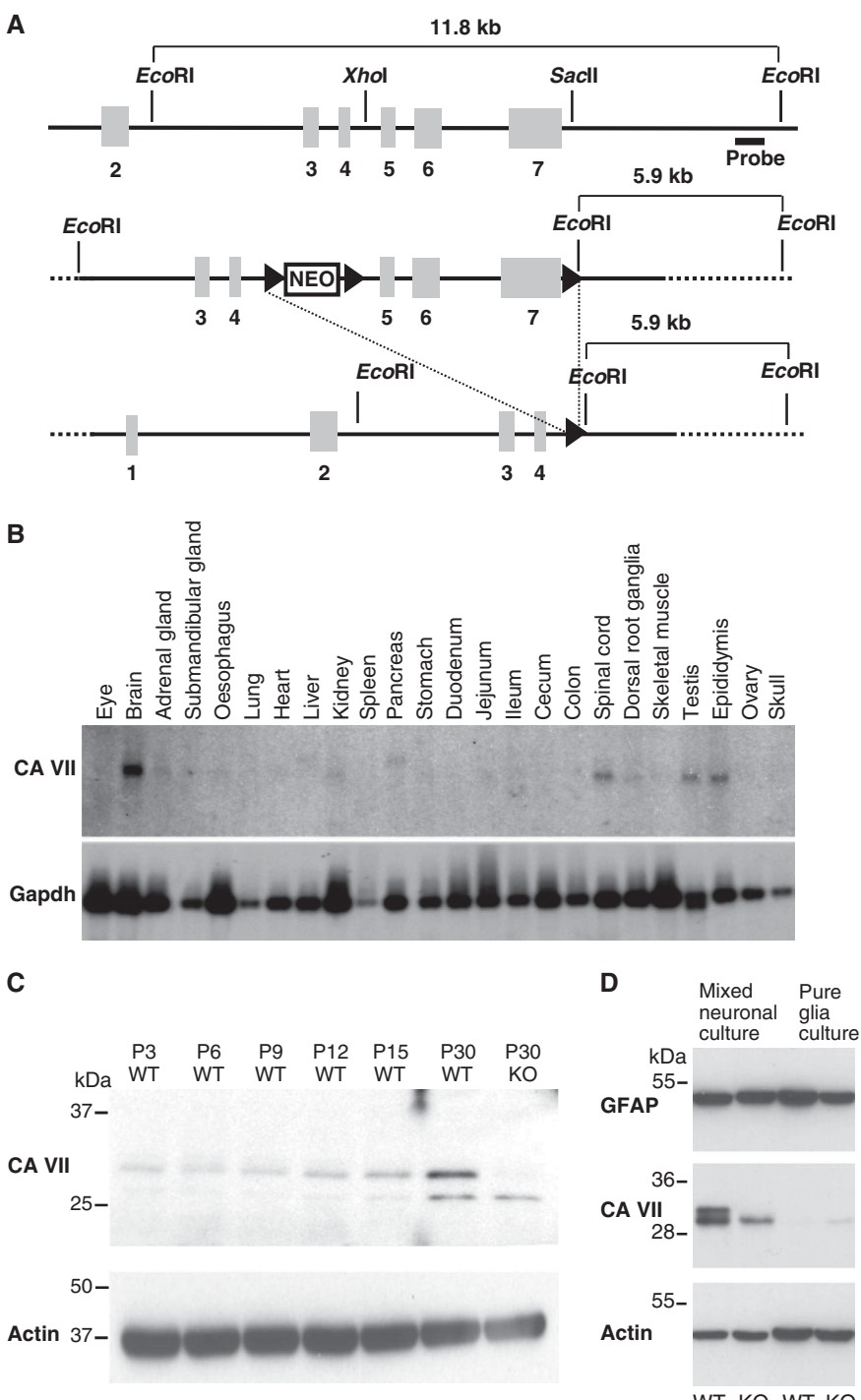

**Figure 1** Generation of CA VII-deficient mice. (**A**) Targeting strategy for the *Car7* locus. Endogenous *Car7* locus (top). Targeting construct (middle). Targeted locus after Cre-mediated recombination (bottom). Exons are indicated as grey boxes and loxP sites as arrowheads. (**B**) Multiple tissue northern blot for CA VII. In the P56 mouse, CA VII is prominently expressed in the central nervous system including brain and spinal cord. *Gapdh* served as the loading control. (**C**) Developmental profile of CA VII protein expression. Western blot analysis of protein lysates of whole hippocampi shows increasing CA VII levels from P3 to P30. The band of the appropriate size ($\sim 30$ kDa) was absent in protein lysates from KO tissue. (**D**) In western blots from protein lysates of cultured cells, CA VII was detected in mixed glia/neuron cultures but not in glia cell cultures. The presence of astrocytes in all cultures was confirmed by detection of GFAP. Actin served as the loading control in **C** and **D**.

responses were evoked with high-frequency stimulation applied at the border of *stratum radiatum (sr)* and *stratum lacunosum-moleculare (slm)* in the presence of ionotropic glutamate receptor and $GABA_BR$ antagonists. In both WT and CA VII KO neurons, this stimulation evoked a biphasic response consisting of an initial hyperpolarization followed

by a prolonged depolarization, both of which were abolished by picrotoxin (Figure 3A). The gradual positive shift has been shown to closely reflect the $HCO_3^-$-driven conductive net uptake of $Cl^-$ (Viitanen *et al*, 2010). Importantly, the slope of the depolarizing shift measured under current clamp conditions was considerably slower in CA VII KO than in

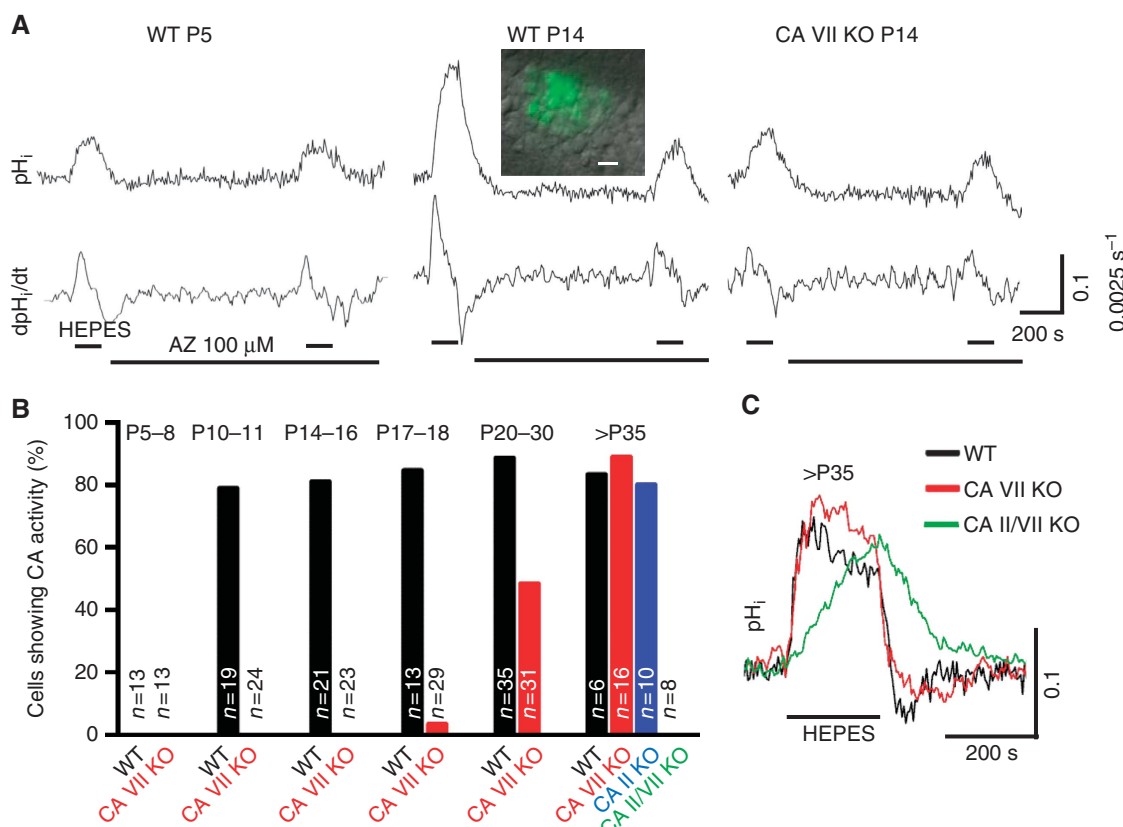

**Figure 2** Development of cytosolic CA activity in mouse CA1 pyramidal neurons is based on sequential expression of isoforms VII and II. (**A**) Original single-cell $pH_i$ traces and their time derivatives from P5 and P14 WT and P14 CA VII KO neurons (baseline $pH_i$ 7.21, 7.13 and 7.11, respectively). Superfusion with $CO_2/HCO_3^-$-free HEPES solution (upper horizontal bars) evoked an intracellular alkalinization, which in P14 WT was large and suppressed by 100 μM acetazolamide (AZ, lower horizontal bar), indicating the presence of CA activity (see Supplementary Materials and methods). The possible effect of AZ on extracellular CAs was excluded by adding 10 μM benzolamide (a poorly permeant CA inhibitor). Inset shows an overlay of the BCECF fluorescence signal and Dodt gradient contrast image of CA1 pyramidal neurons in P14 WT (scale bar 10 μm). (**B**) Summary of the results obtained using the cytosolic CA activity detection method shown in A and quantified as the percentage of cells showing cytosolic CA activity. Data from CA II KO and CA II/VII KO neurons were obtained only at >P35. The number of cells tested is indicated for each bar. The animal numbers for WT mice at the different age points was $n=3$ (P5–8), $n=2$ (P10–11), $n=3$ (P14–16), $n=2$ (P17–18), $n=2$ (P20–30) and $n=2$ (>P35); for CA VII KO mice the numbers were $n=3$ (P5–8), $n=2$ (P10–11), $n=3$ (P17–18), $n=4$ (P20–30) and $n=2$ (>P35); and for CA II KO and CA II/CA VII KO mice the number was $n=2$ (>P35). (**C**) Typical single pyramidal neuron $pH_i$ responses evoked by withdrawal of $CO_2/HCO_3^-$ (horizontal bar, HEPES) in slices from >P35 WT, CA VII KO and CA II/VII KO mice. The rate of rise and the amplitude of the alkaline shift are identical in WT (P39, baseline $pH_i$ 7.01) and CA VII KO neurons (P40, 7.06), whereas alkalinization develops much more slowly in the CA II/VII KO (P46, 7.04).

WT neurons (Figure 3A). This indicates that the dendritic $HCO_3^-$-dependent net uptake of $Cl^-$ is strongly facilitated by CA activity in P12–P16 WT neurons. Measurements under voltage clamp conditions showed that the apparent charge transfer associated with single inhibitory post-synaptic currents was not different between the two genotypes (Supplementary Figure S3A).

In agreement with previous work on rat slices (Kaila *et al*, 1997; Ruusuvuori *et al*, 2004), the long-lasting GABAergic depolarization was able to induce action potential firing in WT neurons (Figure 3A). In striking contrast to this, action potentials were not observed in any of the 12 P12–16 CA VII KO neurons tested. A confounding factor in these experiments is a possible difference in the intrinsic excitability of WT versus CA VII KO neurons. We examined this by measuring the input resistance and the minimum current injection needed to trigger spiking (rheobase) in whole-cell mode, but no difference was found between neurons from WT and CA VII KO mice (Supplementary Figure S3B).

The role of the two cytosolic CA isoforms in the generation of dendritic depolarizing $GABA_A$ responses was examined

using whole-cell recordings from CA1 pyramidal neurons in WT and CA VII KO slices at P12–16 (Figure 3B). Pressure microinjection of GABA at the border of *sr/slm* gave rise to biphasic membrane potential responses, where the depolarizing phase was much smaller in CA VII KOs than in WT. As expected, the depolarizing phase was selectively blocked in the absence of $CO_2/HCO_3^-$ (in the HEPES buffered-solution).

That GABA is excitatory and induces neuronal firing in a CA VII-dependent manner in intact P12–16 WT neurons was clearly seen in field potential recordings (Figure 3C). In all slices from P12–16 WT animals, pressure injection of GABA at the border of *sr/slm* induced spiking and the number of spikes increased with the duration of the GABA puff (8 ms: $6.9 \pm 2$, 10 ms: $18.0 \pm 3.8$, 12 ms: $22.1 \pm 4.7$, 14 ms: $25.6 \pm 6.0$ and 16 ms: $29.3 \pm 7.8$). Spiking was fully blocked by the $GABA_A$-receptor antagonist SR 95531 (gabazine). In contrast, in slices from CA VII KO mice, GABA application evoked hardly any spiking even with the longest puff.

When examining intracellular GABA responses after P35 using GABA pressure injection as described above, no difference was observed between the WT, CA II KO and CA

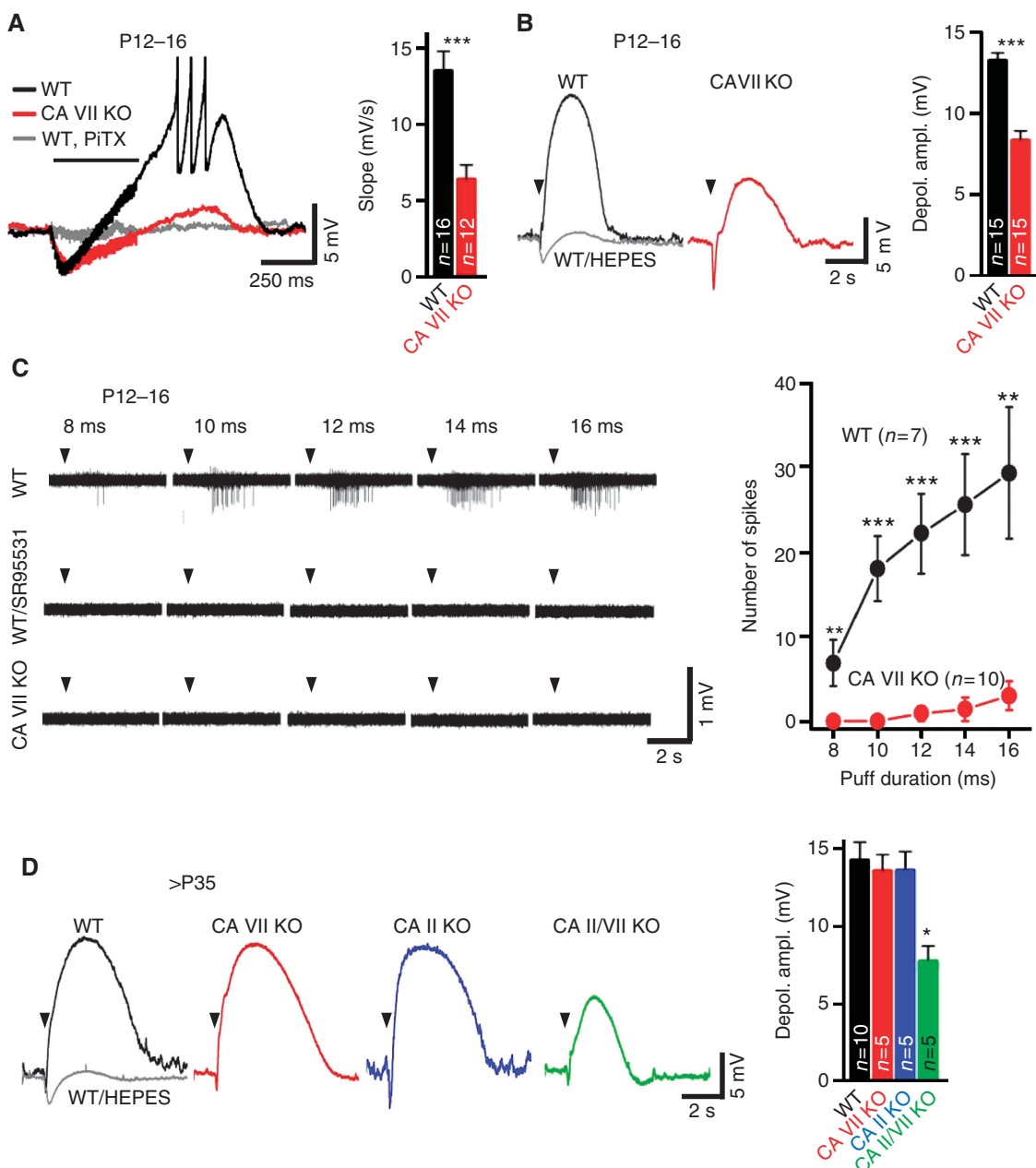

**Figure 3** CA VII and CA II similarly enhance GABA$_A$R-mediated depolarization in CA1 pyramidal neurons. (**A**) Electrical stimulation (40 pulses, 100 Hz at *stratum radiatum/stratum lacunosum-moleculare* (*sr/slm*) border (horizontal bar)) in the presence of AP5/CNQX/CGP55845) evokes an excitatory GABAergic response in WT, but not in CA VII KO neurons. Picrotoxin blocked the biphasic response (PiTX 90 µM; $n=3$/genotype). The mean initial slope of the depolarizing shift was faster in WT than in CA VII KO ($13.53 \pm 1.27$ versus $6.39 \pm 0.96$ mV/s, $P < 0.001$, Student's $t$-test). The resting membrane potential of WT and CA VII KO neurons did not differ ($-71.0 \pm 1.1$ versus $-72.3 \pm 1.4$ mV, respectively; $n=15$ neurons for both genotypes, $P=0.47$, Student's $t$-test). (**B**) Microinjection of GABA (5 mM for 100 ms, *sr/slm* border in the presence of CGP55845/TTX, arrowheads) induced a pronounced depolarization in WT neurons (black) that was abolished in the absence of $CO_2/HCO_3^-$ (grey). The depolarization was smaller in CA VII KO (red). Bar diagram illustrates GABAergic mean peak depolarization in P12–16 WT and CA VII KO ($13.3 \pm 0.5$ versus $8.3 \pm 0.6$ mV, $P < 0.001$, Student's $t$-test). (**C**) In the presence of AP5/CNQX/CGP55845, microinjection of GABA (5 mM, *sr/slm* border) triggered SR 95531-sensitive field potential spikes in WT slices already upon 8-ms injection, while even 16-ms injection evoked hardly any spikes in CA VII KO (left). Mean number of spikes was plotted against GABA puff duration in P12–16 WT and CA VII KO ($P < 0.01$ and $P < 0.001$, Student's $t$-test; right). (**D**) After P35, depolarizations upon GABA microinjection (arrowheads) in the presence of CGP55845/TTX were indistinguishable in WT (black), CA VII KO (red) and CA II KO (blue) but smaller in CA II/VII KO (green). The depolarization in WT was abolished in the absence of $CO_2/HCO_3^-$ (grey, $n=5$). Bar diagram summarizes mean peak depolarization for each genotype (right; ANOVA, Bonferroni). $^*P < 0.05$, $^{**}P < 0.01$ and $^{***}P < 0.001$. Error bars denote s.e.m. and $n$ is the number of slices (**C**) or cells tested (one cell per slice) (**A**, **B** and **D**).

VII KO neurons, indicating that both CA VII and CA II activity can promote depolarizing GABA responses in dendrites (Figure 3D, see also Kaila *et al*, 1997). In agreement with this, the dendritic GABA-evoked depolarization was much smaller in neurons from >P35 CA II/VII double KO animals

(Figure 3D), and similar to that seen in P12–P16 CA VII KO neurons (Figure 3B). The presence of the attenuated but not fully abolished response in the double KO slices is consistent with the fact that, while replenishment of $HCO_3^-$ by CA is rate-limiting for the GABAergic depolarization, the steady-

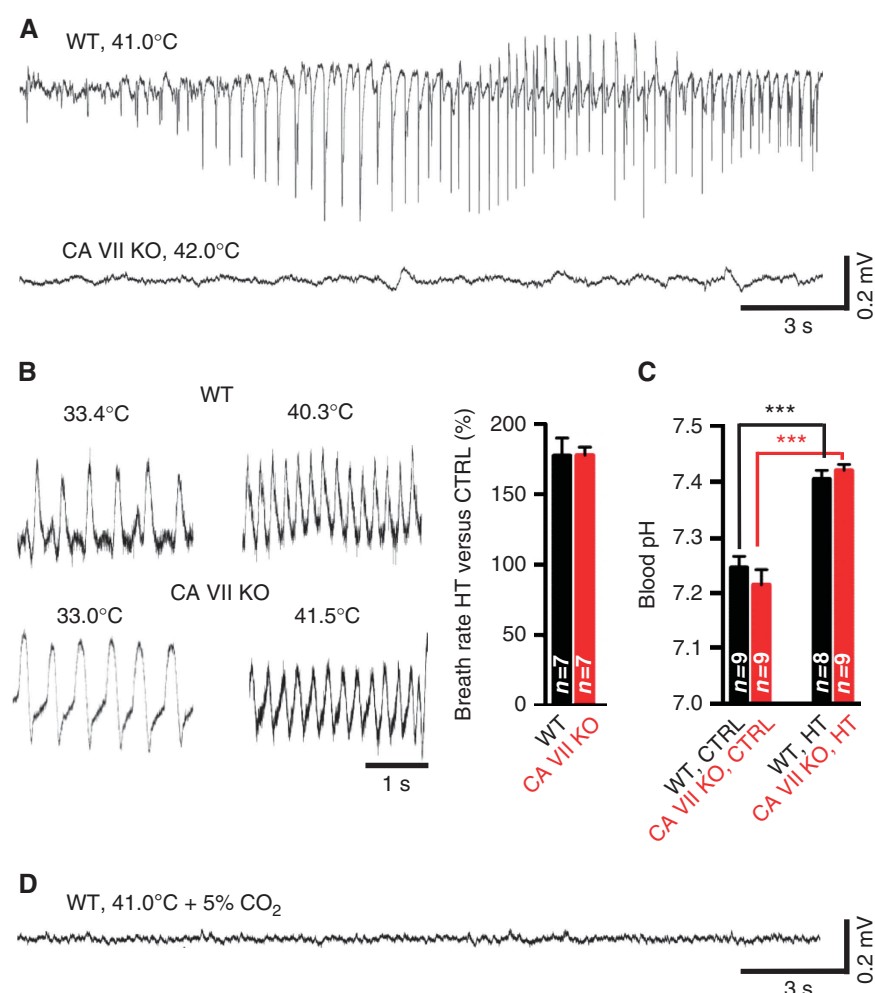

**Figure 4** CA VII promotes experimental febrile seizures in P13–14 mice. (**A**) Representative raw traces of epidural EEG recordings during hyperthermia (HT) from a WT and a CA VII KO mouse. Seizures were reliably induced in WT ($n=9$) but not in CA VII KO ($n=7$). (**B**) Specimen recordings of breath rate from a WT and a CA VII KO mouse. The baseline breath rate (WT $153\pm11$ breaths/min and CA VII KO $157\pm8$ breaths/min) was increased during HT in both genotypes (WT $272\pm28$ breaths/min and CA VII KO $279\pm12$ breaths/min). The bar diagram summarizes the mean increase in breath rate in WT ($177\pm12\%$) and CA VII KO mice ($178\pm6\%$, $P=0.98$). (**C**) Hyperthermia caused a comparable alkalosis in blood pH in both WT and CA VII KO mice. The blood pH did not differ between the two genotypes under control conditions ($P=0.37$) or during HT ($P=0.44$). (**D**) Electrographic seizures were not detected in WT mice when hyperthermia was induced in the continuous presence of 5% $CO_2$ ($n=5$). Error bars denote s.e.m., all $P$-values are based on Student's $t$-test (\*\*\*$P<0.001$), and animal number is given in bar diagrams. Rectal temperatures are given above the traces.

state $HCO_3^-$ concentration of about 10 mM ($pH_i=6.9$, $pH_o=7.3$) is sufficient for a minor depolarizing action.

### Lack of electrographic FS in CA VII KO mice

The seizure susceptibility of a large number of transgenic mice, including knock-in animals with gene variants that promote FS and related epileptiform syndromes in humans, has been tested using eFS triggered by exposure to hyperthermia (Meisler *et al*, 2001; Dube *et al*, 2005; van Gassen *et al*, 2008; Wimmer *et al*, 2010; Hill *et al*, 2011). The distinct developmental expression profiles of CA VII and II made it possible to specifically examine the role of isoform VII in the generation of eFS using P13–14 WT and CA VII KO mice. This developmental stage is considered relevant for comparisons with humans, in whom the peak incidence of FS is reached at the age of 18 months (Berg and Shinnar, 1996; Stafstrom, 2002). eFS were evoked using a pre-heated chamber as described before (Schuchmann *et al*, 2006).

As expected on the basis of previous work (Meisler *et al*, 2001; Dube *et al*, 2005; van Gassen *et al*, 2008), WT mice

($n=4$) when exposed to hyperthermia first showed normal explorative behaviour (score 0 in a behavioural scale of mouse eFS (van Gassen *et al*, 2008)) followed by hyperthermia-induced hyperactivity and escape responses (score 1). Thereafter, immobility and ataxia (score 2) were seen, after which seizures progressed to more severe manifestations including shaking, clonic seizures of one or more limbs (score 3), culminating in continuous tonic-clonic seizures (score 4) at a rectal temperature of $41.9\pm1.1$°C. Remarkably, none of the CA VII KO mice ($n=8$) showed progression of seizures beyond score 3 at similar temperature levels of $42.5\pm0.5$°C. Moreover, the seizure activity in the KO was atypical of eFS also in that it was interspersed with periods during which coordinated behaviour patterns such as walking and exploratory activity were seen, which never occurred in WT mice once seizures had progressed to the clonic stage.

Given the above qualitative difference in the behavioural hyperthermia effects between the WT and CA VII KO, we used EEG to gain more information on the underlying mechanisms.

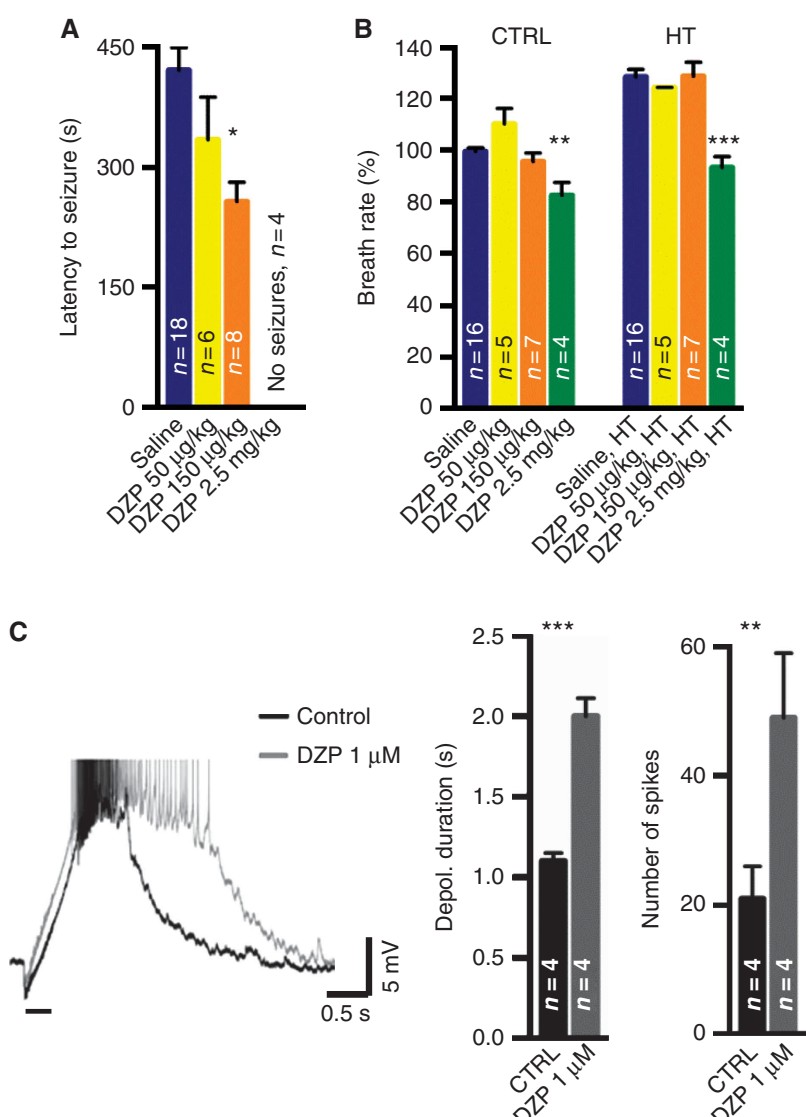

**Figure 5** GABA$_A$-receptor-mediated modulation of experimental febrile seizures. (**A**) Diazepam (DZP, 150 µg/kg) reduced the latency to seizure onset in P14 rat pups ($P<0.05$). There was no statistical difference between 50 µg/kg and saline-injected animals. DZP 2.5 mg/kg completely blocked the seizures. DZP was given intraperitoneally 15 min before hyperthermia (HT) onset. (**B**) Diazepam given at 150 µg/kg did not affect the baseline breath rate, and HT induced an analogous increase in the breath rate in saline- and diazepam-injected rats ($P>0.05$). At a higher concentration (2.5 mg/kg), DZP suppressed breathing significantly ($P<0.01$) and HT caused only a small increase in the breath rate. (**C**) In slice preparation, DZP (1 µM) potentiated the high-frequency stimulation (40 pulses at 100 Hz, horizontal bar) -evoked GABA$_A$R-mediated depolarization. Sample trace from a slice from a P14 rat pup where stimulation to the border of *stratum radiatum* and *stratum lacunosum-moleculare* was given in the presence of AP5, CNQX and CGP 55845. The bar diagrams summarize the effects of DZP on the duration of the depolarization (time to half-maximum) and the number of action potentials associated with the depolarizing phase. *P*-values (*$P<0.05$, **$P<0.01$ and ***$P<0.001$) are based on ANOVA, Bonferroni (**A**, **B**) or Student's *t*-test (**C**), and error bars denote s.e.m. The number of animals (**A**, **B**) or cells (**C**) is indicated in bar diagrams.

Epidural EEG recordings showed that, in nine out of nine WT mice, electrographic seizures commenced within $24\pm4$ min after the onset of hyperthermia (Figure 4A). At this time point, the rectal temperature was $40.9\pm0.4^\circ$C. The seizures were easily recognized as large-amplitude bursts of spikes (150–600 µV) with a frequency of about 2–4 Hz and duration of 10–60 s, which, notably, occurred mainly during periods of behavioural arrest as also reported by others (Dube *et al*, 2000). Because we have previously shown that a respiratory alkalosis is a major trigger of eFS (Schuchmann *et al*, 2006, 2011), we also measured the breath rates as well as blood pH values in the two genotypes. At the time of eFS onset in the WT, the breath rate had increased to $177\pm12\%$ (Figure 4B). In

striking contrast to the WT, no electrographic seizures were observed in seven out of seven CA VII KO animals that were examined (Figure 4A). The rectal temperature ($41.3\pm0.4^\circ$C) and breath rate at 24 min after the onset of hyperthermia in CA VII KO animals were similar to those in WT animals (Figure 4B). Importantly, direct measurements of blood pH showed that hyperthermia induced a respiratory alkalosis with identical magnitude in both genotypes (Figure 4C). These data indicate that a suppression of systemic respiratory alkalosis does not explain the absence of eFS in the CA VII KO mice. Consistent with our earlier findings (Schuchmann *et al*, 2006), the generation of both behavioural and cortical electrographic eFS in WT mice were fully prevented when hyperthermia

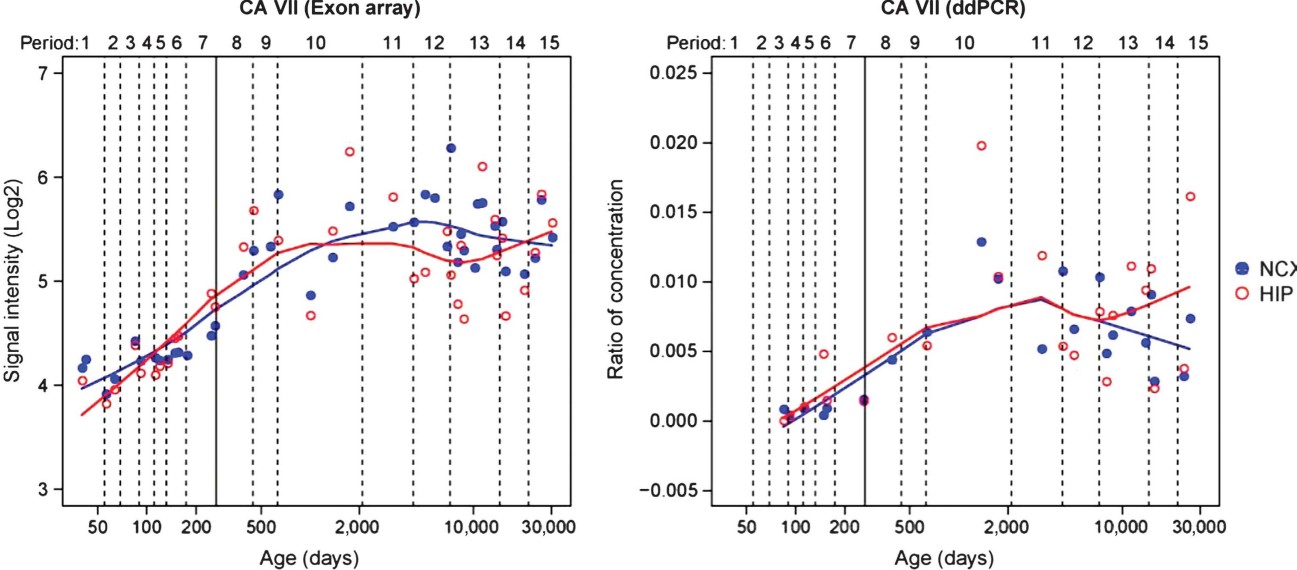

**Figure 6** CA VII expression during human brain development. Line plots show the log$_2$-transformed exon array signal intensity (left), and the normalized ratio of absolute copy number of CA VII to *Gapdh* by droplet digital PCR (ddPCR, right) in the neocortex (NCX) and hippocampus (HIP) from the early fetal period to late adulthood. The solid line between periods 7 and 8 separates the prenatal from the postnatal period.

was induced in the continuous presence of 5% CO$_2$ in air (Figure 4D). Finally, measurements of blood electrolytes showed no genotype-specific differences in control conditions or under hyperthermia (Supplementary Figure S4).

The above data raise the intriguing question whether CA VII-dependent GABAergic excitation contributes to the generation of cortical electrographic eFS. Hence, we performed behavioural experiments on P14 rat pups to examine whether enhancing GABA$_A$R signalling by low doses of diazepam would facilitate the triggering of eFS. When diazepam was given at a very low dose (50 μg/kg intraperitoneally), latency to seizure onset decreased but the difference was not statistically significant from that seen in saline-injected animals (Figure 5A). However, a dose of 150 μg/kg resulted in a significant decrease in seizure latency. Importantly, neither of these two doses had a detectable effect on the baseline breath rate, and hyperthermia was associated with a similar increase in breath rate in rats injected with 50–150 μg/kg and in those injected with saline (Figure 5B). As could be expected (Hirtz and Nelson, 1983; Laorden *et al*, 1990), a higher dose of diazepam (2.5 mg/kg) prevented seizure generation in four of four rats tested, while the saline-injected control animals showed robust eFS (Figure 5A). At this concentration, diazepam suppressed the breath rate under control conditions and, under hyperthermia, only a modest increase in breathing was seen in the presence of the drug (Figure 5B).

In order to further examine the hypothesis that CA VII-dependent GABAergic excitation facilitates the generation of eFS, we evoked pharmacologically isolated excitatory GABAergic responses in P14 rat hippocampal slices. Consistent with the above hypothesis, the depolarizing phase as well as spiking evoked by the high-frequency stimulation was considerably enhanced by application of 1 μM diazepam (Figure 5C), with a maximum effect seen after 10 min of drug application.

As will be discussed below, the above *in vivo* and *in vitro* data as a whole suggest that eFS are facilitated by GABAergic excitation and that a low concentration of diazepam potentiates

these effects by a selective action on GABA$_A$Rs (Eghbali *et al*, 1997). The suppression of eFS by a high dose of diazepam is readily explained by the suppression of breathing (Figure 5B).

### Spatiotemporal expression patterns of human CA VII in the developing brain

We used exon array analysis and droplet digital PCR to examine the developmental expression of CA VII in the human neocortex and hippocampus (see Supplementary Materials and methods). Our previous study on the human brain transcriptome identified 29 modules corresponding to specific spatiotemporal expression patterns by weighted gene co-expression network analysis (Kang *et al*, 2011). Here, it is of much interest that, among them, CA VII was classified as a member of M2 (Supplementary Figure S5), a main module including 2745 genes, which is interpreted as a module related to synaptic transmission based on functional annotations using gene ontology (Kang *et al*, 2011). Our present data show that CA VII expression is markedly increased in the human neocortex and hippocampus during the perinatal period and sustained until late adulthood (Figure 6). Thus, an association between CA VII and epileptiform syndromes is not excluded at any stage of human neocortical and hippocampal development.

## Discussion

Distinct CA isoforms exert a profound modulatory influence on the functions of a wide variety of voltage-gated channels, excitatory and inhibitory synapses and gap junctions (see Introduction). Because isoform-specific CA inhibitors are not available, we used gene disruption of two cytosolic isoforms, CA II and CA VII, separately and in combination to demonstrate the following: (i) CA activity is fully attributable to these two isoforms in the somata and dendrites of mature mouse CA1 pyramidal neurons; (ii) the two isoforms are sequentially expressed in the developing hippocampus, with expression of CA VII commencing at P10 and that of CA II at

around P20; (iii) in contrast to CA II, CA VII is not expressed in glia; (iv) both isoforms are able to promote $HCO_3^-$-dependent GABAergic depolarization and excitation triggered by intense $GABA_A$R activation in mature neurons. This kind of information is crucial for understanding the mechanisms of pH-dependent modulation of neuronal and neuronal network functions in the developing and mature brain (Kaila and Ransom, 1998; Schuchmann *et al*, 2009; Casey *et al*, 2010; Tolner *et al*, 2011; Magnotta *et al*, 2012). We also show here that (v) excitatory CA VII-dependent GABA signalling promotes eFS at P14, and the relevance of this finding is underscored by the observation (vi) that CA VII is expressed in the human cortex perinatally, well in advance of the age of 6 months when FS are first seen in children (Berg and Shinnar, 1996; Stafstrom, 2002).

Our previous work on rat hippocampal slices has demonstrated that the emergence of excitatory $HCO_3^-$-dependent $GABA_A$R-mediated responses in CA1 neurons (Kaila *et al*, 1997) coincides with the onset of CA VII mRNA expression at around P10–12 (Ruusuvuori *et al*, 2004). However, these data demonstrate a correlation but not a cause–effect link between CA VII expression and GABAergic excitation because the possibility remained that some other, unidentified isoform might have been expressed in parallel with CA VII. The present data, which are based on the use of a novel CA VII KO mouse as well as a CA II KO (Lewis *et al*, 1988) and a CA II/CA VII double KO, clearly indicate that the two isoforms are sequentially expressed. CA VII is the only cytosolic CA isoform during P10–P18, while the two isoforms are co-expressed starting at around P20. Given the steeply increasing interest in the physiology, pathophysiology and pharmacology of individual CA isoforms (Supuran, 2008), this kind of information is of immense value in the design and evaluation of experiments on neuronal CAs, and makes the hippocampal developmental time window of $\sim$P12–16 (see Figure 2) eminently suited for examining the properties of CA VII in isolation. This isoform is interesting in that it is mainly expressed in the brain; moreover, in brain parenchyma its expression is restricted to neurons. The ubiquitous CA II isoform is expressed in both neurons and glia and abundantly outside the CNS (Ghandour *et al*, 1980; Wang *et al*, 2002; Supuran, 2008). Notably, the complete absence of CA activity in pyramidal neurons before P10 and in CA II/CA VII double KO neurons at all ages clearly shows that mature pyramidal neurons express only these two cytosolic isoforms.

### Somatodendritic developmental expression of CA II and CA VII in pyramidal neurons

As demonstrated by the present somatic $pH_i$ recordings and excitatory dendritic $GABA_A$R responses, the subcellular somatodendritic expression patterns of CA VII and CA II do not differ from each other. Moreover, both isoforms were equally effective in promoting somatic $CO_2$-dependent pH shifts separately from, and in combination with, each other. The intrinsic catalytic rates of CA II and CA VII are extremely high (Earnhardt *et al*, 1998), and hence it is obvious that the present assay (see Figure 2) does not reflect the kinetic properties of these two isoforms. Nevertheless, our data strongly suggest that neither isoform acting in isolation would be rate-limiting for $pH_i$ changes during major physiological and pathophysiological acid-base shifts.

A possible explanation for the presence of two cytosolic CA isoforms after P20 is that they show differences in their biochemical functions other than (de)hydration of $CO_2$, such as esterase/phosphatase activity (Innocenti *et al*, 2008). Furthermore, the $HCO_3^-$ transporters make a significant contribution to neuronal pH regulation (Hentschke *et al*, 2006; Jacobs *et al*, 2008; Sinning *et al*, 2011). By forming isoform-specific metabolons with distinct acid-base transporters, CA II and VII may contribute to developmentally and spatially distinct $pH_i$ microdomains (Sterling *et al*, 2001; Becker and Deitmer, 2007; see also Boron, 2010; Stridh *et al*, 2012). Moreover, recent work suggests that CA VII acts as scavenger of oxygen radicals (Truppo *et al*, 2012), which fits well with the large increase in cortical ocidateve energy metabolism taking place at the time of onset of CA VII expression (Erecinska *et al*, 2004).

### GABAergic excitation is promoted by both CA VII and CA II

Work on single crayfish muscle fibres and neurons originally demonstrated that the net efflux of intracellular $HCO_3^-$ across $GABA_A$Rs leads to GABAergic depolarization by exerting a direct depolarizing action and, indirectly, by facilitating conductive uptake of $Cl^-$ (Kaila and Voipio, 1987; Kaila *et al*, 1989; Voipio *et al*, 1991). The $GABA_A$R-mediated CA-dependent acidosis is readily explained by the fact that $HCO_3^-$ is a substrate of cytosolic CAs, and a conductive net loss of $HCO_3^-$ will by necessity shift the balance of the CA-catalysed (de)hydration reaction to more acidic $pH_i$ values (Kaila *et al*, 1990; Luckermann *et al*, 1997). Accordingly, in mammalian neurons and especially in their dendrites, which have a much larger surface-to-volume ratio (Qian and Sejnowski, 1990), CA activity is generally thought to be necessary for maintaining the supply of intracellular $HCO_3^-$ that drives GABAergic depolarizing and excitatory responses (Pasternack *et al*, 1993; Viitanen *et al*, 2010). In the present work, we addressed the roles of the two isoforms in the generation of $HCO_3^-$-dependent GABAergic depolarization. Our data show that CA VII has a unique role in promoting $Cl^-$ accumulation (Viitanen *et al*, 2010) and consequent depolarizing GABAergic responses during P10–18, a time window that is known to be associated with a heightened proneness to epileptogenesis in rodents (Swann *et al*, 1993; Gloveli *et al*, 1995). The co-expression of isoforms VII and II starting at P20 had no obvious effect on the $GABA_A$R-mediated depolarization.

### Role of CA VII-dependent excitatory GABA transmission in the generation of eFS

The molecular and neuronal network mechanisms underlying eFS have been extensively studied in various inbred and mutant mouse strains at the age of P14 (van Gassen *et al*, 2008; Wimmer *et al*, 2010). This is a very intriguing time point in the present context, because it enables a comparison of susceptibility to eFS of mice that have, or are completely devoid of, cytosolic CA in pyramidal neurons—that is, the WT and CA VII KO mice, respectively.

We have previously shown that a respiratory alkalosis promotes the generation of eFS in rodents (Schuchmann *et al*, 2006), and a recent retrospective clinical study has shown that a respiratory alkalosis is characteristic of FS in children (Schuchmann *et al*, 2011). Notably, the present data on breath rates and direct measurements of blood pH show

that CA VII plays no role in hyperthermia-induced respiratory alkalosis in rodents. Here, one should recall that the non-catalysed (de)hydration of $CO_2$ has a half-time of 8 s (Geers and Gros, 2000) at a typical physiological temperature. CA activity is therefore not likely to be important for the extracellular and consequent intracellular pH changes (cf. Ruusuvuori et al, 2010) that are associated with the slow hyperthermia-induced respiratory alkalosis, which has a time-to-seizure induction of about 20 min. In contrast, intraneuronal CA activity is needed for the fast replenishment of $HCO_3^-$ and consequent net uptake of $Cl^-$, which are key mechanisms in the generation of excitatory $HCO_3^-$-dependent GABAergic responses (Ruusuvuori et al, 2004; Viitanen et al, 2010). Indeed, the present EEG data indicate that CA VII plays a central role in the generation of electrographic seizure activity associated with eFS in the cortex of P14 mice. The hypothesis that CA VII facilitates eFS by boosting GABAergic excitation is supported by two lines of evidence: (i) the behavioural eFS in P14 rats were enhanced by a low dose of diazepam, which is likely to act solely on $GABA_A$Rs (Eghbali et al, 1997), while a high concentration blocked seizures; and (ii) parallel in vitro experiments showed that the excitatory GABAergic response was markedly potentiated by 1 μM diazepam. Interestingly, a recent paper has demonstrated an enhancement of GABAergic transmission in the GABRG2$^{R43/Q43}$ knock-in mouse model of human FS (Hill et al, 2011).

The above data point to a facilitatory role for CA VII activity and consequent GABAergic excitation in the generation of FS. Importantly, the difference between genotypes in eFS susceptibility cannot be attributed to a difference in body temperature at the onset of the seizures. The dual effects of diazepam in vivo raise the intriguing possibility (see also Schuchmann et al, 2011) that the high therapeutic concentrations needed for suppression of FS in children exert their action via suppression of breathing rather than acting directly on the neuronal networks that are immediately responsible for the generation of FS. Indeed, it is a well-known fact that high concentrations of diazepines interfere with respiratory functions, especially in children (Orr et al, 1991; Tasker, 1998; Norris et al, 1999; Appleton et al, 2008). The expected outcome is block of the respiratory alkalosis with consequent suppression of FS.

Importantly, the expression of CA VII in the human cortex and hippocampus, as demonstrated in the exon array and droplet digital PCR analyses, already starts perinatally, while FS are typically seen in children at 6 months of age with the incidence peaking at 1.5 years (Berg and Shinnar, 1996; Stafstrom, 2002). CA inhibitors have been used for decades as anticonvulsant agents, but their side effects, including interference with bone formation (Lehenkari et al, 1998), preclude their long-term use in children. Notably, CA VII shows an expression profile that is mainly neuronal. Thus, designing isoform-specific inhibitors of CA VII has much potential as a novel approach in the treatment of FS and possibly of other epileptiform syndromes.

## Materials and methods

Animal experiments were approved by the responsible local institutions and complied with the regulations of the US National Institutes of Health and with those of the Society for Neuroscience (USA). Researchers were blind to the genotype during experiments and data analysis. Human tissues were collected as before (Kang et al, 2011) after obtaining parental or next-of-kin consent and with approval by the relevant institutional review boards.

### Generation of CA VII KO mice
The Car7 gene was disrupted by homologous recombination in ES cells and blastocyst injection as detailed in Results and in Supplementary Materials and methods.

### Northern analysis
Northern blot analysis was performed with RNA isolated from various tissues of adult mouse as described previously (Jacobs et al, 2008). The probe corresponded to nucleotides 210–492 of the Car7 reference sequence (NM_053070.3). Gapdh served as a loading control.

### Antibody generation and western analysis
The CA VII antiserum was raised in rabbits against the epitope DNFRPPQPLKGRVVK (amino acids 245–259) of the CA VII protein (accession number NM_053070.3) coupled via an N-terminal cysteine to keyhole limpet haemocyanin and affinity-purified. For western blots shown in Figure 1, 10 μg protein lysates from whole hippocampi or from cultured cells was separated on reducing 12% SDS–polyacrylamide gels and blotted on to a PVDF membrane as described in Sinning et al, 2011. Blots were probed with the rabbit CA VII antibody at a dilution of 1:250. Detection was performed with the chemiluminescence ECL Kit (Amersham Biosciences). Comparison of immunostainings for CA VII on brain sections from WT and KO mice revealed that our antibody was not suitable for immunolocalization studies because of unspecific staining. Mixed neuron/glia cultures or glial cell cultures were performed as described previously (Boettger et al, 2003). For additional western blots, see Supplementary Materials and methods.

### Generation of CA II/VII double KO mice
Animals heterozygous for both CA II and CA VII alleles were used to generate CA II/VII double KO mice. For more details on the CA II/VII double KO and genotyping, see Supplementary Materials and methods.

### Electrophysiology and pH$_i$ fluorescence imaging
For details on electrophysiological recordings and BCECF fluorescence imaging of pH$_i$ from hippocampal CA1 pyramidal neurons, see Supplementary Materials and methods.

### EEG recordings and eFS induction
In P14 rat pups, eFS were induced using a pre-heated chamber as described elsewhere (Schuchmann et al, 2006). For intracranial EEG recordings and for detection of behavioural hyperthermia effects in P13–14 mice, the temperature of the chamber was set to $43 \pm 1°C$. For details see Supplementary Materials and methods.

### Blood analysis
GEM 4000 (Instrumentation Laboratory, Bedford, MA) was used in the analysis of trunk blood electrolytes and pH. Arteriovenous blood (80 μl) from control mice and from mice that underwent hyperthermia was collected after decapitation.

### CA VII detection in developing human brain
Exon array analysis was performed as previously described (Kang et al, 2011). For secondary validation using droplet digital PCR analysis, an aliquot of the total RNA that was previously extracted from each brain region was used. For more details, see Supplementary Materials and methods.

### Statistical analysis
All numerical data are expressed as mean ± s.e.m. Statistical analyses of data were performed with two-tailed Student's t-test or with ANOVA, Bonferroni as appropriate. P-values are given in the figure legends.

### Supplementary data
Supplementary data are available at The EMBO Journal Online (http://www.embojournal.org).

## Acknowledgements

This study was supported by grants from the Academy of Finland, the Sigrid Jusélius Foundation, and the Jane and Aatos Erkko Foundation to KK, by grants from the DFG to CAH and by grant from NIMH (MH081896) to NS. KK is a member of the Finnish Center of Excellence in Molecular and Integrative Neuroscience Research.

*Author contributions*: ER, AKH, IK, HJK, JV, NS, CAH and KK designed the research; AKH, IK, PB, MH, AYY, HJK, MEM and JCH performed the research; ER, AKH, IK, PB, MH, AYY, HJK, MEM, JCH, JV, NS, CAH and KK analysed the data; and ER, IK, HJK, JV, NS, CAH and KK wrote the paper.

## Conflict of interest

The authors declare that they have no conflict of interest.

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
