## [Review Process File · The EMBO Journal]

Manuscript EMBO-2013-84363

Neuronal carbonic anhydrase VII provides GABAergic excitatory drive to exacerbate febrile seizures

Eva Ruusuvuori, Antje K Huebner, Ilya Kirilkin, Alexey Yukin, Peter Blaesse, Mohamed Helmy, Hyo Jung Kang, Malek El Muayed, J Christopher Hennings, Juha Voipio, Nenad Sestan, Christian A Hübner and Kai Kaila

Corresponding author: Kai Kaila, University of Helsinki

Review timeline:

Submission date:	09 January 2013
Editorial Decision:	04 March 2013
Revision received:	28 May 2013
Editorial Decision:	18 June 2013
Revision received:	20 June 2013
Accepted:	20 June 2013

Transaction Report:

Editor: Karin Dumstrei

1st Editorial Decision

04 March 2013

Thank you for submitting your manuscript to the EMBO Journal. Your study has now been reviewed by three referees and their comments are provided below.

As you can see, the referees find the analysis interesting and suitable for publication in the EMBO Journal. They raise a number of specific points that should be resolved before further consideration here. Given the positive comments provided, I would like to invite you to submit a revised version of the manuscript, addressing the comments of all three reviewers. I should add that it is EMBO Journal policy to allow only a single round of revision and that it is therefore important to address the concerns at this stage.

When preparing your letter of response to the referees' comments, please bear in mind that this will form part of the Review Process File, and will therefore be available online to the community. For more details on our Transparent Editorial Process, please visit our website: <http://www.nature.com/emboj/about/process.html>

Thank you for the opportunity to consider your work for publication. I look forward to your

revision.

REFEREE REPORTS

Referee #1

The authors report on changes in susceptibility to high body temperature induced seizures in mice which lack two different carboanhydrases. Their data is valuable as it shows that CA expression is developmentally regulated and that it is not only CAVII which is the only CA in central neurons as previously suggested by the Kaila group. By demonstrating developmental time course and sites of expression they provide for useful information not only with respect to cellular pH Homeostasis but also with respect to other functions of bicarbonate which relate to chloride concentration to mitochondrial function and to Ca homeostasis.

They suggest that their data indicates enhanced depolarization by GABA being involved in generation of febrile seizures. The authors expand on previous studies which show that alkalosis in young animals is critical for seizure induction.

This paper addresses the potential role which GABA may play. Gaba A receptor channels are permeable to bicarbonate as first shown by Bormann and colleagues in the Sakmann lab causing depolarization of neurons if the membrane potential and the chloride equilibrium potential are identical as would be the case when cells are exposed to either tonic inhibition by GABA or prolonged and increased release of GABA. The reversal potential for bicarbonate is around minus 20 mV. This is set by the capability of neurons by the carboanhydrase to produce bicarbonate from CO₂ generated in the conversions of oxygen and glucose to CO₂ and water for production of ATP. Lack of a carboanhydrase would reduce the intracellular HCO₃ concentration and thereby shift the equilibrium potential of bicarbonate in hyperpolarizing direction. Consequently animals which lack CA should be less excitable and agents which facilitate GABA receptor activation such as benzodiazepines or barbiturates should become proconvulsant. This is in essence what this paper tries to show and this is certainly an important contribution to the reason why hyperthermia may cause seizures.

Critique

1. The authors like others term these seizures febrile seizures although they are caused by hyperthermia and are in the strict sense of definition not febrile as the classical PGE etc signaling cascade in shifting the temperature homeostasis are not involved in hyperthermia. Mice and rats lack regulation against warming by environment being the reason that they choose environments where sweating is not required. Their only chance to control for temperature is therefore hyperventilation causing alkalosis. This reason does not fully transfer to children where also electrolyte loss is critically involved. Febrile seizures in babies come usually about when the infection is due to pulmonary disease and not when gastrointestinal infections dominate. Hyperthermia seizures on the other hand are not limited to babies.

2. A simple test for the hypothesis promoted by the authors would be application of muscimole or other subunit specific GABA agonists. If they are right muscimole should induce seizures in the relevant age groups. This is to my knowledge not the case.

3. Previous reasons for hyperexcitability by alkalosis have addressed the effects on surface charge screening, on NMDA receptors and on H currents. This paper does not discriminate which of these factors are more important.

4. In infants seizure susceptibility is generally increased. This applies to the neocortex the entorhinal cortex and the hippocampus and is true for different seizure induction protocols such as fluothyl, 4-AP low Mg, low Ca etc). This applies also to humans and may relate to febrile seizures particularly. In this study hyperthermia is used to induce seizures which works also in adult humans. There the seizure induction is also related to loss of Ca and Mg as occurs also in eclampsia. It would be helpful to know what are the changes in electrolytes in plasma at the time of seizure onset particularly as in young age due to incomplete "blood brain barrier" such changes readily transfer to the interstitial space. In slices from bay rats relatively small changes in Ca and Mg concentration are

known to induces seizures.

5. The authors perform their study mostly in area CA1 of the hippocampus which is only under very specific conditions a seizure onset site due to its high packing density and low extracellular space volume fraction. This does not occur in human tissue. It would have been good to know where seizure start and do the study in such a region. One condition in which CA1 is particularly seizure prone is the high K model in Wistar rats (not Sprague Dawley) and the low Ca model (not in human hippocampus) and another aspect is development where CA1 was shown previously to participate in seizures generated in entorhinal cortex in age P10 to P20. One such study is for example Gloveli et al 1995 Dev Brain Res and another by Weissinger et al in Neurobiology of disease 2000).

6. In humans application of CO₂ seems to work also in adults in early status epilepticus as was taught in emergency medicine courses in the early 60ties in some countries.

Specific other comments:

1. In infants seizure susceptibility is generally increased. In a recent study the role of depolarizing GABA with respect to acetazolamide on one hand and bumetanide on the other hand was addressed (Wahab et al Epilepsia 2010). The study shows that susceptibility to such treatments is different in different regions of the brain as well and that there are time periods where acetazolamide does not work yet which is compatible with the present study. In area CA1 the effect of prolonged GABA was studied by application of THDOC which prolongs IPSPs and causes shift from hyperpolarizing to depolarizing without increased excitability (Burg Heinemann and Schmitz Eur J Neuroscience 1998). This suggests that other factors may contribute such as immature K homeostasis, increased expression of NMDA receptors and the like.

2. The sentence starting in abstract with " A low dose of diazepam" is unclear. Does it relate to animals lacking CA II or CAVII or both?

3. It would be nice to know the pattern of expression in neocortex and entorhinal cortex. In Ec particularly as it is likely in the same slides as the hippocampus and is more seizure prone than the hippocampus proper.

4. From the change in cytosolic pH the change in bicarbonate concentration could be estimated and put in relation to bicarbonate reversal potential and its change when acetazolamide is applied.

5. Whole cell patch recordings are not necessarily the best way to study changes in equilibrium potential as the cells are exposed to changes in Donnan equilibration etc. Perforated patches would have been more appropriate or sharp microelectrode recordings.

6. In a study by Romo Para and Gutierrez depolarizing GABA was detected in apical dendrites up to P 20 or more and this was not dependent on bicarbonate suggesting that expression of NKCC1 is compartment dependent. How would such a complication affect the data of this study?

7. Local application of GABA causes changes in extracellular slow field potential. If GABA is depolarizing the fp change would be negative and vice versa. Did the authors check this as it would support their interpretation?

8. The intrinsic excitability was determined using a rheobase protocol for determining changes in CA deficient mice. Bicarbonate is also a good Ca buffer (the reason why following Mody and colleagues people apply 1.5 - 1.6 mM Ca in order to achieve a Ca concentration equivalent to 1.2 as found in most animal species so far). If less of the entering Ca is buffered there should be more activation of Ca dependent K currents but in the long term also effects on mitochondrial ATP synthesis. Hence information should be provided on I/Fr curves and the like in which different depolarizing steps are used to evaluate cellular excitability. Such info is more realistic in perforated patch or sharp microelectrode recordings.

9. Application of GABA in different states of development can also be modulated by changes in GABA uptake (see for example Draguhn and others). This is why the Gutierrez group preferred muscimol for their study of local sensitivity to GABA. Another way would have been differential

stimulation in presence of glutamate receptor antagonists which readily induces monophasic IPSPs and which are different in properties dependent where the stimulation is applied.

10. The data on human cortex are pretty premature and lack post mortem control as available by comparing surgical specimen and postmortem tissue.

Referee #2

Understanding mechanisms which underlie febrile seizures early in development is critical for the successful treatment of this condition. The paper by Ruusuvuori et al. extends the previous research performed by K. Kaila's group. This study uses new lines of knockout mice to address the role of two forms of carbonic anhydrase (CA) in the GABA(A)R-mediated excitation during febrile seizures in the young mice. The authors show that the expression of CA VII precedes that of CA II and its activity contributes to the excitatory action of excessive GABA(A)R activation thereby exacerbating experimental febrile seizures. In general, the experiments are carefully done, the paper is sound and well written, and conclusions are justified.

I have several relatively minor comments:

- 1) The experimental procedure of the pH_i measurements should be described in more details. The inset in fig. 2A shows a fluorescent smear which is not confined to the boundaries of neurons. How long BCECF-AM was applied/washed-out before the imaging and how the contribution of the extracellular fluorescence was excluded? It is also unclear what criteria were set to distinguish between neurons that did or did not show CA activity. Judging by the traces shown on the figure, AZ also decreased, although to a lesser extent, the fluorescence signal in WT P5 and CA VII KO P14 animals. Is this related to photobleaching? The stability of the readout signal should be evaluated. A proper quantification of the results would also be welcomed.
- 2) Please, provide the number of animals used in each age group in experiments reported in fig. 2B.
- 3) It is a bit surprising that despite CA VII KO mice show behavioural signs of seizures up to score 3 with shaking and clonic limb seizures this was not reflected in the EEG. Can authors comment on that?
- 4) It would be more reassuring if the experiments where GABA(A)R-mediated excitation is compared in the WT and CA VII KO P14 mice were performed using less invasive techniques, e.g. in cell-attached or perforated patch mode.
- 5) Depolarizing effects of GABA puffs would depend on the number of activated GABA(A)Rs/the surface of the dendritic tree subjected to the saturating GABA concentrations. It is therefore important to ensure that these parameters are similar in all experiments. Given that pressure injections of GABA were performed at a certain distance from the somata of recorded neurons it is not at all clear how this has been controlled.
- 6) I am puzzled by the fact that the breath rate increase of almost 80% during hyperthermia in WT mice (fig. 4B) is not replicated in the diazepam experiments, where all groups, including saline injected animals, displayed only about 30% rise (fig. 5B). What is the reason for this discrepancy?
- 7) The paper suggests that the reduced seizure susceptibility of the CA VII KO P14 mice is due to the less pronounced GABA(A)R-mediated depolarization resulting from the lack of CA activity at this age. Do CA blockers have anti-ictogenic effects in this model? Could the authors demonstrate that the effects of diazepam on excitatory GABA(A) responses are abolished by AZ?
- 8) The authors suggest that excitatory CA-dependent GABA signaling promotes neonatal seizures. If CA VII is expressed and functional in humans at birth why febrile seizures only start to occur at 6 months of age? Does this not indirectly point towards the role of later expressed CA II as well?

Referee #3

This is an interesting, well-written, and highly relevant manuscript that uncovers the developmental expression sequence of carbonic anhydrase 7 (CA7) at the protein level, and uses single and dual knock out strategies, to link CA7 and CA2 to GABA-A receptor-mediated depolarizing responses, that follow prolonged, high frequency activation of interneuron afferents. Parallel studies of febrile seizure in wild type vs. CA7 KO mice implicate this CA isoform and GABA-A receptors in this pathology.

The experiments are well described and the data clearly presented. This is an excellent study and an important contribution of general neurobiological and neurological relevance.

Major Comments - none.

Minor Comments.

1. Figure 2: quantitative comparisons of ΔpHi and dpH/dt are valid with the assumption that the intrinsic buffering capacity of the neurons is unchanged. A rather drastic change in buffering would be required, and therefore seems unlikely but the point (albeit minor) remains.

2. The authors may wish to comment on how the 40 pulse, 100 Hz train (used to elicit depolarizing GABA-A responses) relates to afferent inhibitory input a pyramidal cell would experience in the run up to a febrile seizure. Would such prolonged, high frequency bursts of interneurons indeed occur, or is the point more in the line of general reasoning that some increasing degree of inhibitory compromise is to be expected in the run up, owing to the catalyzed replenishment of bicarbonate efflux?

3. Microinjection of GABA caused a smaller and slower depolarization in the double KO (Fig. 3C) presumably due to the steady state concentration of cytosolic bicarbonate. Also, in the CA7 KO, there was a depolarization, due to 100 Hz stimulation for 40 s, albeit smaller and slower. These observations raise the question of the role of CA7-catalyzed HCO_3^- formation in normal, fast GABAergic inhibitory synaptic transmission. Inhibitory interneurons such as basket cells will typically fire a high frequency bursts. Does CA7 play a role in this context?

1st Revision - authors' response

28 May 2013

Referee #1

The authors report on changes in susceptibility to high body temperature induced seizures in mice which lack two different carboanhydrases. Their data is valuable as it shows that CA expression is developmentally regulated and that it is not only CAVII which is the only CA in central neurons as previously suggested by the Kaila group. By demonstrating developmental time course and sites of expression they provide for useful information not only with respect to cellular pH Homeostasis but also with respect to other functions of bicarbonate which relate to chloride concentration to mitochondrial function and to Ca homeostasis.

They suggest that their data indicates enhanced depolarization by GABA being involved in generation of febrile seizures. The authors expand on previous studies which show that alkalosis in young animals is critical for seizure induction.

This paper addresses the potential role which GABA may play. Gaba A receptor channels are permeable to bicarbonate as first shown by Bormann and colleagues in the Sakmann lab causing depolarization of neurons if the membrane potential and the chloride equilibrium potential are identical as would be the case when cells are exposed to either tonic inhibition by GABA or prolonged and increased release of GABA. The reversal potential for bicarbonate is around minus 20 mV. This is set by the capability of neurons by the carboanhydrase to produce bicarbonate from CO_2 generated in the conversions of oxygen and glucose to CO_2 and water for production of ATP. Lack of a carboanhydrase would reduce the intracellular HCO_3^- concentration and thereby shift the equilibrium potential of bicarbonate in hyperpolarizing direction. Consequently animals which lack CA should be less excitable and agents which facilitate GABA receptor activation such as benzodiazepines or barbiturates should become proconvulsant. This is in essence what this paper tries to show and this is certainly an important contribution to the reason why hyperthermia may cause seizures.

We don't think the referee expects a response to these introductory notes.

Critique

1. The authors like others term these seizures febrile seizures although they are caused by

hyperthermia and are in the strict sense of definition not febrile as the classical PgE etc signaling cascade in shifting the temperature homeostasis are not involved in hyperthermia. Mice and rats lack regulation against warming by environment being the reason that they choose environments where sweating is not required. Their only chance to control for temperature is therefore hyperventilation causing alkalosis. This reason does not fully transfer to children where also electrolyte loss is critically involved.

- We'd like to emphasize that in our present and previous work, we have been very careful to call the seizures observed in the rat pup model "experimental febrile seizures" (eFS) to make the relevant distinction emphasized to by the Referee. The usage of the term eFS has also been explained in our review (Schuchmann, Vanhatalo and Kaila BrainDev 2009). At the end of the Abstract we use the abbreviation FS because of the implications to the human condition referred to in the previous sentence.

Febrile seizures in babies come usually about when the infection is due to pulmonary disease and not when gastrointestinal infections dominate. Hyperthermia seizures on the other hand are not limited to babies.

- Here, it might be appropriate to point out (for the editor's attention) that the senior author of the present manuscript is also the senior author of a recent study on the role of respiratory alkalosis in FS generation in children with pulmonary disease and gastrointestinal infections (Schuchmann et al. Epilepsia 2011). The rat pup model we use is designed to mimic, as closely as possible, some of the key aspects of febrile seizures as described in the original paper (Schuchmann et al./Kaila Nature Medicine 2006; see also review by Schuchmann et al./Kaila, Brain & Development, 2009). The mechanisms underlying hyperthermia seizures in adults are beyond the scope of our present and previous work.

2. A simple test for the hypothesis promoted by the authors would be application of muscimole or other subunit specific GABA agonists. If they are right muscimole should induce seizures in the relevant age groups. This is to my knowledge not the case.

- We do not understand this suggestion. What we show is that low concentrations of diazepam, a positive allosteric modulator of GABA_A receptors, enhances eFS. Interestingly, in vitro experiments showed a potentiation of synaptic depolarizing GABA_A responses by this drug which is known to preferably target subsynaptic GABA_A receptors. Such an effect is not mimicked by muscimol or any GABA_A receptor agonist. There is absolutely nothing in our explanatory framework that would predict that muscimol application as such would lead to seizures. We also show that high concentrations of diazepam suppress the eFS in parallel with a suppression of breathing, which leads to the intriguing idea that the therapeutic action of diazepam on FS in children might be mediated by suppression of breathing.

3. Previous reasons for hyperexcitability by alkalosis have addressed the effects on surface charge screening, on NMDA receptors and on H currents. This paper does not discriminate which of these factors are more important.

- We agree that there are numerous ligand- and voltage-gated channels (including also some GABA_ARs, Ca²⁺ channels, K⁺ channels, ASICs etc) as well as gap junctions which are sensitive to pH changes. Our article contains references to several papers on these topics. Changes in surface charges are also relevant and may play a role in pH modulation of ion-channel properties. However, our mechanistic analyses are focused on the roles of intraneuronal CA isoforms and GABA_AR signaling, and studying all possible pH-modulatory mechanisms that have an influence on neuronal excitability goes clearly beyond the aims and scope of the present study. In fact, such a broad question would be impossible to address in any single experimental project. But we have now added two references (Hamon B, Stanton PK, and Heinemann U (1987) *Neurosci Lett*, and Hille B (2001) p. 653-654) to cite the relevant literature even more extensively. As also noted by the Referee, a major and important conclusion of our paper is that depolarizing GABA actions boosted by CA VII facilitate eFS.

4. In infants seizure susceptibility is generally increased. This applies to the neocortex the entorhinal cortex and the hippocampus and is true for different seizure induction protocols such as

fluothyl, 4-AP low Mg, low Ca etc). This applies also to humans and may relate to febrile seizures particularly. In this study hyperthermia is used to induce seizures which works also in adult humans. There the seizure induction is also related to loss of Ca and Mg as occurs also in eclampsia. It would be helpful to know what are the changes in electrolytes in plasma at the time of seizure onset particularly as in young age due to incomplete "blood brain barrier" such changes readily transfer to the interstitial space. In slices from bay rats relatively small changes in Ca and Mg concentration are known to induce seizures.

- Electrolytes: eFS in the rat pup hyperthermia model can be induced within 3 minutes (work by T.Z. Baram and many other labs) and we have shown that also in such a fast-heating model, hyperventilation is a key feature (Tolner et al. Epilepsia 2011). This makes it highly unlikely if not impossible that eFS are triggered by changes in blood electrolytes. In order to minimize heat-induced pain that is bound to take place in the "Baram model", we use a more gentle hyperthermia procedure where seizures commence within about 30 minutes. Nevertheless, we have now carried out experiments where plasma electrolytes have been measured in WT and CA VII KO mice under control conditions and after hyperthermia. These data show that plasma Ca²⁺, Na⁺ and Cl⁻ levels are identical in both genotypes under control conditions. A small reduction is seen in Ca²⁺ concentration in hyperthermia but this change is identical in WT and CA VII KO mice and therefore cannot explain the lack of seizures in the latter. The data are shown in the ms as Supplementary Figure S4.
- Blood-brain barrier: The idea that the BBB in a P6 rat would be "leaky" has never gained direct experimental support. In fact, according to direct measurements by Butt et al. (JPhys 1990) and more recent ones done in our lab, the BBB of a P6 rat is not leaky as shown by experiments on sodium fluorescein extravasation into the brain and by the electrophysiological characteristics of the BBB (Helmy et al./Kaila, Brain 2012).
- Slices: We agree that very small changes in (divalent) ion concentrations have an effect on the excitability of slices. However, such data do not translate into in vivo conditions as is evident from the above considerations.

5. The authors perform their study mostly in area CA1 of the hippocampus which is only under very specific conditions a seizure onset site due to its high packing density and low extracellular space volume fraction. This does not occur in human tissue. It would have been good to know where seizure start and do the study in such a region. One condition in which CA1 is particularly seizure prone is the high K model in Wistar rats (not Sprague Dawley) and the low Ca model (not in human hippocampus) and another aspect is development where CA1 was shown previously to participate in seizures generated in entorhinal cortex in age P10 to P20. One such study is for example Gloveli et al 1995 Dev Brain res and another by Weissinger et al in Neurobiology of disease 2000).

- Several teams have tried to locate the site of origin of eFS but so far it has not been identified. In particular, whether the entorhinal cortex plays a key role is not known. Our in vitro data are from CA1 because the depolarizing GABAergic responses have been previously characterized in detail in this brain area (Kaila et al., 1997 JNS; Smirnov et al., 1999 JNS, Ruusuvoori et al., 2004 JNS; Viitanen et al., 2010 JPhysiol). CA1 is, of course, a major output pathway from the hippocampus and therefore likely to be relevant in seizure spread. In addition, the role of depolarizing and excitatory GABA_AR-mediated responses have been examined in CA1 area in most published papers dealing with CA actions on neuronal excitability.
- The former of the above two papers based on slice work might provide a link to the putatively convulsant GABAergic K⁺ transient analyzed in Viitanen et al/Voipio 2010, and we have in fact cited Gloveli et al. in the original version of the ms (p. 14).

6. In humans application of CO2 seems to work also in adults in early status epilepticus as was taught in emergency medicine causes in the early 60ties in some countries.

- This is very interesting in view of our lab's focus on CO₂ actions, but unfortunately we cannot cite this statement as such in our present paper.

Specific other comments:

1. In infants seizure susceptibility is generally increased. In a recent study the role of depolarizing GABA with respect to acetazolamide on one hand and bumetanide on the other hand was addressed

(Wahab et al Epilepsia 2010). The study shows that susceptibility to such treatments is different in different regions of the brain as well and that there are time periods where acetazolamide does not work yet which is compatible with the present study. In area CA1 the effect of prolonged GABA was studied by application of THDOC which prolongs IPSPs and causes shift from hyperpolarizing to depolarizing without increased excitability (Burg Heinemann and Schmitz Eur J Neuroscience 1998) . This suggests that other factors may contribute such as immature K homeostasis, increased expression of NMDA receptors and the like.

- The senior author of the present work is familiar with all the above papers from the Heinemann lab. Unfortunately, using the CA VII KO we have found that acetazolamide has profound off-target effects (unpublished data) thereby modifying neuronal network activity in a CA-independent manner. Thus, the drug is suitable as a specific tool only in experiments of the kind shown in Fig. 2 where CA functionality can be directly monitored. This is one of the reasons that the present work was based on a KO strategy.

- Yes, we of course do agree that seizure generation depends on a multitude of molecular and cellular mechanisms. The aim of our work is to identify a novel mechanism involved in eFS generation (depolarizing GABA signaling) which may have consequences in translational and clinical work on human FS.

2. The sentence starting in abstract with "A low dose of diazepam" is unclear. Does it relate to animals lacking CA II or CAVII or both?

- eFS studied in this work are monitored at P13-14, at a time when CA VII is expressed in the rodent hippocampus but CA II is absent. We agree that the sentence is ambiguous and should be modified to read "A low dose of diazepam promotes eFS in P13-P14 rat pups, whereas seizures are blocked at concentrations that suppress breathing". As is evident from the Results, we decided to use rat pups in these experiments because of the large data basis we have on behavioral eFS in rats; and because behavioral seizures are much more reliable for quantitative analysis in rats – an observation made by a number of teams working on eFS and other seizure paradigms (see response to Referee 2, point 3).

3. It would be nice to know the pattern of expression in neocortex and entorhinal cortex. In Ec particularly as it is likely in the same slides as the hippocampus and is more seizure prone than the hippocampus proper.

- Please see our response to point #5 above.

4. From the change in cytosolic pH the change in bicarbonate concentration could be estimated and put in relation to bicarbonate reversal potential and its change when acetazolamide is applied.

- In this study we use acetazolamide only when probing intraneuronal carbonic anhydrase activity using fluorescent imaging of cytoplasmic pH. We do not quite understand how changes in E(HCO₃) relate to this paradigm. The role of HCO₃ in setting E_{GABA} has been discussed extensively in our previous original papers and reviews (for reviews: Kaila, 1994; Farrant and Kaila, 2007).

5. Whole cell patch recordings are not necessarily the best way to study changes in equilibrium potential as the cells are exposed to changes in Donnan equilibration etc. Perforated patches would have been more appropriate or sharp microelectrode recordings.

- In our previous papers, we present a wealth of data that the depolarizing GABA response evoked by high-frequency stimulation in adult neurons looks identical in whole-cell as well as in sharp-microelectrode recordings indicating a largely dendritic origin (Kaila et al., 1997 JNS; n numbers for both types of recording about 50). However, in our experience sharp electrodes are not suitable for juvenile mouse neurons which have a rather high input resistance. Because the functionally excitatory nature of the depolarizing GABA action is of crucial importance in the present work, we have added novel data on spiking in intact neurons, evoked by dendritic GABA puffs and monitored using field-potential recordings. In excellent agreement with our working hypothesis, exogenous application of GABA evoked vigorous neuronal spiking in slices from P12-P16 WT mice but almost none in the P12-P16 KO (which is devoid of cytosolic CA) (Figure 3 C).

6. In a study by Romo Para and Gutierrez depolarizing GABA was detected in apical dendrites up

to P 20 ore more and this was not dependent on bicarbonate suggesting that expression of NKCC1 is compartment dependent. How would such a complication affect the data of this study?

- We do not claim that all kinds of depolarizing GABAergic responses share an identical mechanistic basis. Indeed, a role for NKCC1 has been demonstrated in several types of juvenile and adult neurons (Blaesse et al., Neuron 2009). In our present work, a role for NKCC1 is ruled out by the fact that the depolarizing responses are strictly dependent on the presence of bicarbonate (see Figs 3 B and D).

7. Local application of GABA causes changes in extracellular slow field potential. If GABA is depolarizing the fp change would be negative and vice versa. Did the authors check this as it would support their interpretation?

- Please see our response to point #5 above.

8. The intrinsic excitability was determined using a rheobase protocol for determining changes in CA deficient mice. Bicarbonate is also a good Ca buffer (the reason why following Mody and colleagues people apply 1.5 - 1.6 mM Ca in order to achieve a Ca concentration equivalent to 1.2 as found in most animal species so far). If less of the entering Ca is buffered there should be more activation of Ca dependent K currents but in the long term also effects on mitochondrial ATP synthesis. Hence information should be provided on I/FR curves and the like in which different depolarizing steps are used to evaluate cellular excitability. Such info is more realistic in perforated patch or sharp microelectrode recordings.

- We agree that Ca²⁺ buffering substances including bicarbonate reduce Ca²⁺ activity in solutions that have a given total Ca²⁺ concentration, and that Ca²⁺ buffering in the cytosol attenuates Ca²⁺ transients and affects their dynamics. However, cellular Ca²⁺ regulation sets the level of intracellular Ca²⁺ activity, not the total Ca²⁺ content of a cell. Therefore, it is quite speculative to assume that a hypothetical CA- and bicarbonate-dependent change in intraneuronal Ca²⁺ buffering would cause a shift in the resting Ca²⁺ level and lead to altered excitability via Ca²⁺ dependent conductances and/or energy metabolism.

- Importantly, novel data on cellular excitability (field recordings of GABA-puff evoked spiking) are provided in the revised ms.

9. Application of GABA in different states of development can also be modulated by changes in GABA uptake (see for example Draguhn and others). This is why the Gutierrez group preferred muscimole for their study of local sensitivity to GABA. Another way would have been differential stimulation in presence of glutamate receptor antagonists which readily induces monophasic IPSPs and which are different in properties dependent where the stimulation is applied.

- We have studied the depolarizing response using HFS in the presence of glutamate and GABA_BR blockers (Fig 3 A) and using GABA application in the presence of GABA_B blocker and TTX (Fig 3 B-D). Please note also that we are comparing different genotypes within an identical developmental time window, which rules out developmental stage as a source of variability in our phenotypic characterization.

10. The data on human cortex are pretty premature and lack post mortem control as available by comparing surgical specimen and postmortem tissue.

- The data are from the same material as in the milestone paper by Kang et al/Sestan (Nature 2012) which describes the developmental transcriptome of the human brain.

However, we do appreciate the reviewer's concerns as the use of post-mortem human tissue could potentially alter gene and protein expressions due to ante- and post-mortem conditions.

Unfortunately, due to ethical and practical limitations in using human post-mortem material this is not possible to completely avoid. The use of surgical specimen does not constitute a proper control as this tissue is removed because of an underlying brain disease, most often epilepsy, and usually from patients that have been medicated for a considerable length of time. Importantly, human specimens used in this study were obtained from clinically unremarkable donors and with short PMI. Furthermore, the specimens were genotyped to exclude genomic abnormalities and extensively evaluated by a neuropathologist to assess for hypoxia, cerebrovascular incidents, tumors, microbial

infections, neurodegeneration and demyelination. Thus, we have taken the necessary measures to ensure that the postmortem specimens used in this study are of high quality, similar to the material used in the Kang et al./Sestan Nature paper referred to above.

Referee #2

Understanding mechanisms which underlie febrile seizures early in development is critical for the successful treatment of this condition. The paper by Ruusuvuori et al. extends the previous research performed by K. Kaila's group. This study uses new lines of knockout mice to address the role of two forms of carbonic anhydrase (CA) in the GABA(A)R-mediated excitation during febrile seizures in the young mice. The authors show that the expression of CA VII precedes that of CA II and its activity contributes to the excitatory action of excessive GABA(A)R activation thereby exacerbating experimental febrile seizures. In general, the experiments are carefully done, the paper is sound and well written, and conclusions are justified.

I have several relatively minor comments:

1) The experimental procedure of the pHi measurements should be described in more details. The inset in fig. 2A shows a fluorescent smear which is not confined to the boundaries of neurons. How long BCECF-AM was applied/washed-out before the imaging and how the contribution of the extracellular fluorescence was excluded? It is also unclear what criteria were set to distinguish between neurons that did or did not show CA activity. Judging by the traces shown on the figure, AZ also decreased, although to a lesser extent, the fluorescence signal in WT P5 and CA VII KO P14 animals. Is this related to photobleaching? The stability of the readout signal should be evaluated. A proper quantification of the results would also be welcomed.

- The "smear" is caused by out-of-focus neurons. The ROIs (diameter about 3x5 μm) we use are positioned well within a brightly-fluorescent neuronal soma.
- In ratiometric measurements with reasonable S/N, photobleaching should not contribute to the changes in the kinetics of the fluorescence signal. In Ruusuvuori et al. 2004 we reported a small reduction in the dpHi/dt at P0-8 in the presence and also in the absence of EZA when $\text{CO}_2/\text{HCO}_3^-$ withdrawal was repeated under control conditions twice or more. Hence, there seems to be a rundown effect in the repetitive alkaloses. This kind of minor rundown is often seen under various types of pHi manipulations using a weak acid/base to evoke intracellular pH transients, but it is obvious that they do not have an influence on our quantifications (see Methods). Any possible effect of extracellular CA activity was eliminated using 10 μM benzolamide in the perfusion solution.
- The criteria were set conservatively, i.e. at 2x SD (cf. Ruusuvuori et al., 2004). This would at maximum lead to a 1-day error delay in the detection of CA activity as is also clear from the distribution of values in graph 2B.
- We do not agree that, taking into account the S/N ratio of the primary recordings, there is a detectable effect of AZ in the WT P5 and CA VII KO P14 animals in Fig. 2. Indeed, we feel that if we would claim that there IS such an effect, any time-series statistics applied to these traces would give non-significant differences with regard to wash-in of AZ.
- A quantification based on the criteria explained above is given in the plot in Fig. 2B. We do not see what additional quantification would be needed.

2) Please, provide the number of animals used in each age group in experiments reported in fig. 2B.

- The animal numbers are included in the figure legend of the revised ms.

3) It is a bit surprising that despite CA VII KO mice show behavioural signs of seizures up to score 3 with shaking and clonic limb seizures this was not reflected in the EEG. Can authors comment on that?

- Electroclinical dissociation in video-EEG monitoring of mice is a well-known and frequent phenomenon. In addition, there are lots of data in mice on behavioral "freezing" associated with electrographic cortical seizures during eFS. Our work strongly suggests that CA VII is a key molecule in the generation of cortical electrographic seizures, but has not a similarly dramatic effect on seizure semiology.

4) *It would be more reassuring if the experiments where GABA(A)R-mediated excitation is compared in the WT and CA VII KO P14 mice were performed using less invasive techniques, e.g. in cell-attached or perforated patch mode.*

We agree that less invasive techniques have advantages when studying consequences of transmembrane Cl⁻ fluxes in the perisomatic region. However, it is well known that diffusion-based control of concentrations is effective and fast only within very small distances (for one-dimensional diffusion, the diffusion delay increases proportional to the square of distance; see e.g. the last few lines on page 313 in Bertil Hille's monography (2001) "Ion Channels of Excitable Membranes, Third Edition", or page 17 in Albert Einstein's "Investigations on the theory of the Brownian movement", Dover Publications Inc., 1956, translation from the 1905 original: "...in the direction of the X-axis... mean displacement is therefore proportional to the square root of the time..."). In the present study, we have addressed dendritic CA activity by evoking GABA_AR mediated Cl⁻ transients in distal dendrites that are far enough from the somatic site of patching to make perfusion via the pipette ineffective against Cl⁻ transients that have a time scale of seconds. However, since this question was pointed to also by Referee 1, we used field potential recordings combined with local microinjections of GABA to provide novel, noninvasive data. Please see also our response to this kind of comment by Referee 1 (point 5).

5) *Depolarizing effects of GABA puffs would depend on the number of activated GABA(A)Rs/the surface of the dendritic tree subjected to the saturating GABA concentrations. It is therefore important to ensure that these parameters are similar in all experiments. Given that pressure injections of GABA were performed at a certain distance from the somata of recorded neurons it is not at all clear how this has been controlled.*

- The data speak for themselves and the parameters for pressure injection were controlled as follows. The amount of GABA injected was kept as constant as possible. The apical dendrites of CA1 pyramidal neurons were followed using Dodt gradient contrast image starting from the site of recording to the border of stratum radiatum/lacunosum-moleculare, and the puff pipette was placed in the superficial layer of the slice in s.l.m. immediately behind the s.r./s.l.m. border. Positioning of the puff electrode is now more clearly described in the Supplementary Methods. Accurate positioning is the only way (to our knowledge) whereby reproducible results can be obtained from experiments of this kind. Our lab has a very long track-record in these types of experiments.

6) *I am puzzled by the fact that the breath rate increase of almost 80% during hyperthermia in WT mice (fig. 4B) is not replicated in the diazepam experiments, where all groups, including saline injected animals, displayed only about 30% rise (fig. 5B). What is the reason for this discrepancy?*

- Please note that the comparison made above by the Referee is between data on mice and rats. The increase in breath rate under hyperthermia is lower in rats than in mice, but they fit very well as "interpolated" age dependent data points into our previous observations of rats recorded at P8-P12 and ~P20 (Schuchmann et al./Kaila, Nat Med 2006).

7) *The paper suggests that the reduced seizure susceptibility of the CA VII KO P14 mice is due to the less pronounced GABA(A)R-mediated depolarization resulting from the lack of CA activity at this age. Do CA blockers have anti-ictogenic effects in this model? Could the authors demonstrate that the effects of diazepam on excitatory GABA(A) responses are abolished by AZ?*

- Unfortunately there are no CA isoform-specific inhibitors available. Acetazolamide is a broad-spectrum CA inhibitor that at this age would block several CA isoforms both in the brain (e.g CA II in glia and choroid plexus and the extracellularly catalytic CA IV and CA XIV in brain tissue) and elsewhere (e.g. kidney and lungs). The systemic acidosis associated with AZ is well documented and an acidosis itself has seizure-suppressing effects in many different seizure models (Tolner et al., 2011; Schuchmann et al, 2006). Hence, the results of AZ on eFS would be hard to interpret in terms of "reduced CA VII-dependent GABA depolarization". Also, as pointed out in the response to Referee 1 (specific comment #1), AZ has turned out to be a "dirty" compound. This is in fact typical for amines, and even after consulting leading workers in the CA antagonist field (e.g. Claudiu Supuran), we have no drug available that could be safely used for an effect of the above kind. These are the main reasons why we chose to use the tedious KO approach in our mechanistic studies.

8) The authors suggest that excitatory CA-dependent GABA signaling promotes neonatal seizures. If CA VII is expressed and functional in humans at birth why febrile seizures only start to occur at 6 months of age? Does this not indirectly point towards the role of later expressed CA II as well?

- Expression of CA VII is a *necessary* but (obviously) not a *sufficient condition* for the generation of electrographic eFS. We would expect that a major change takes place in the wiring of the human brain at about 6 months which is necessary for the generation of FS. This is also a developmental stage where the human EEG undergoes a number of qualitative changes, which adds to the evidence for a macroscopic change in cortical circuitries (not just presence or absence of distinct neuronal CA isoforms) at around 6 months of age. CA II is expressed in glia and in neurons and, taken together, the above facts make us reluctant to speculate on a possible role of CA II in human FS.

Referee #3

This is an interesting, well-written, and highly relevant manuscript that uncovers the developmental expression sequence of carbonic anhydrase 7 (CA7) at the protein level, and uses single and dual knock out strategies, to link CA7 and CA2 to GABA-A receptor-mediated depolarizing responses, that follow prolonged, high frequency activation of interneuron afferents. Parallel studies of febrile seizure in wild type vs. CA7 KO mice implicate this CA isoform and GABA-A receptors in this pathology.

The experiments are well described and the data clearly presented. This is an excellent study and an important contribution of general neurobiological and neurological relevance.

Major Comments - none.

Minor Comments.

1. Figure 2: quantitative comparisons of delta pHi and dpH/dt are valid with the assumption that the intrinsic buffering capacity of the neurons is unchanged. A rather drastic change in buffering would be required, and therefore seems unlikely but the point (albeit minor) remains.

- We agree, of course, with the Referee that quantitative pH responses are affected by changes in the intrinsic buffering capacity. However, our comparisons are based on the effects of the presence vs absence of the CO₂/HCO₃ buffer and cytosolic functional CA as identified by a CA inhibitor in individual neurons. Thus, our conclusions regarding the presence/absence of cytosolic CA activity are not affected by changes in intrinsic buffering power.

2. The authors may wish to comment on how the 40 pulse, 100 Hz train (used to elicit depolarizing GABA-A responses) relates to afferent inhibitory input a pyramidal cell would experience in the run up to a febrile seizure. Would such prolonged, high frequency bursts of interneurons indeed occur, or is the point more in the line of general reasoning that some increasing degree of inhibitory compromise is to be expected in the run up, owing to the catalyzed replenishment of bicarbonate efflux?

- Our aim with the high-frequency pulse train is simply to mimic the situation where a neuron is the target of a massive GABAergic barrage, such as is known to occur via feedforward interneurons during epileptic seizures (see e.g. work by A. Trevelyan and collaborators). And, indeed, in the presence but not absence of functional cytosolic CA, the GABAergic barrage leads to excitation in the CA1 target neurons.

3. Microinjection of GABA caused a smaller and slower depolarization in the double KO (Fig. 3C) presumably due to the steady state concentration of cytosolic bicarbonate. Also, in the CA7 KO, there was a depolarization, due to 100 Hz stimulation for 40 s, albeit smaller and slower. These observations raise the question of the role of CA7-catalyzed HCO₃⁻ formation in normal, fast GABAergic inhibitory synaptic transmission. Inhibitory interneurons such as basket cells will typically fire a high frequency bursts. Does CA7 play a role in this context?

- As we have shown before, GABAergic excitation is a population response (Kaila et al., 1997; Viitanen et al., 2010) which is based on the concerted activity of a large number of interneurons and a large extracellular K⁺ transient. Whether high-frequency bursts of individual interneurons might cause substantial ionic shifts in their target cells (or subcellular compartments thereof) is not known but this is certainly an interesting question. In the present work, examining synchronous population responses is motivated, given the link to seizures as set out in the aims of our work.

2nd Editorial Decision

18 June 2013

Thank you for submitting your revised manuscript to the EMBO journal. Your study has now been re-reviewed by referees #1 and 2. As you can see below both referees appreciate the introduced changes and support publication here. Referee #1 has a few minor remaining comments that can be addressed with appropriate text changes. No new experiments are needed. Once we get the revised manuscript back we will proceed with the acceptance of the paper for publication here.

Thank you for submitting your interesting manuscript to the EMBO Journal.

REFEREE REPORTS

Referee #1

This is a revised manuscript which addresses most of the previous criticism in a satisfactory form

I have some minor queries

1. Did Diazepam really suppress respiration or just reduce respiration and was the effect due to anoxia, acidosis or increased GABA function
2. The authors might consider that in neurons which express tonic GABA currents ECl is ion those cells close to RMP due to the large conductance as shown by the Mody lab and by others. In such neurons the effects of HCO₃⁻ gradient on E GABA would be particularly large
3. In a study by Burg et al GABAergic inhibition in CA1 was increased by application of a neurosteroid which lead to depolarizing stimulus evoked GABA responses under physiological conditions. Such measurements are possible using perforated patch recordings or sharp micro electrode recordings. The problem with focal Gaba application is that the duration of GABA presence is much longer than occurring under physiological conditions saturating KCC2 transport capacity. The interesting point is that some of the neurosteroids are now developed as anticonvulsants. The same would apply to Tiagabine which block GABA uptake and thereby enhances GABAergic function. Testing or at least discussing both possibilities would strengthen the paper or recording from cells which are equipped with tonic GABA currents such as DG granule cells which have anyhow depolarizing GABA responses (see Misgeld and Klee and recent confirmation by Jonas and colleagues.
4. The quoted Hamon paper does not address the issue of alkalosis and surface charge screening although it deals with surface charge screening to some extent. If the authors want to quote papers from the Heinemann lab there is one which might be of interest in the context of this study: Velíšek L, Dreier JP, Stanton PK, Heinemann U, Moshé SL. Lowering of extracellular pH suppresses low-Mg(2+)-induces seizures in combined entorhinal cortex-hippocampal slices.

Referee #2

The authors have adequately addressed my comments, and I am satisfied with the revised version of the manuscript.

2nd Revision - authors' response

20 June 2013

Responses to the Referee comments

We are happy that both referees are satisfied with the revisions done in the manuscript.

Referee #1

This is a revised manuscript which addresses most of the previous criticism in a satisfactory form I have some minor queries

1. Did Diazepam really suppress respiration or just reduce respiration and was the effect due to anoxia, acidosis or increased GABA function

- The data show a clear cause-effect whereby diazepam suppresses breathing. "Reduction" would perhaps also be an appropriate expression but we are using standard terminology here. We don't see how (in light of all the data provided) it would be possible to explain the suppression of breathing by anoxia or acidosis. We find that the data shown exclude these possibilities.

2. The authors might consider that in neurons which express tonic GABA currents E_{Cl} is ion those cells close to RMP due to the large conductance as shown by the Mody lab and by others. In such neurons the effects of HCO_3^- gradient on E_{GABA} would be particularly large.

- This is an interesting point which has also been discussed in our team and in a number of our oral presentations on GABA actions. However, there is no obvious link to the present study and discussing this topic would be out-of-focus in our paper.

3. In a study by Burg et al GABAergic inhibition in CA1 was increased by application of a neurosteroid which lead to depolarizing stimulus evoked GABA responses under physiological conditions. Such measurements are possible using perforated patch recordings or sharp micro electrode recordings. The problem with focal Gaba application is that the duration of GABA presence is much longer than occurring under physiological conditions saturating KCC2 transport capacity. The interesting point is that some of the neurosteroids are now developed as anticonvulsants. The same would apply to Tiagabine which block GABA uptake and thereby enhances GABAergic function. Testing or at least discussing both possibilities would strengthen the paper or recording from cells which are equipped with tonic GABA currents such as DG granule cells which have anyhow depolarizing GABA responses (see Misgeld and Klee and recent confirmation by Jonas and colleagues.

- The Referee suggests conducting two entirely novel series of experiments, one on steroid modulation of GABA_ARs and one on modulation of GABA uptake. While these topics are of general interest in e.g. studies on anticonvulsant mechanisms, we also find that these experiments are beyond the scope of the present study and would prefer not to discuss these issues in our paper.

4. The quoted Hamon paper does not address the issue of alkalosis and surface charge screening although it deals with surface charge screening to some extent. If the authors want to quote papers from the Heinemann lab there is one which might be of interest in the context of this study: Velisek L, Dreier JP, Stanton PK, Heinemann U, Moshé SL. Lowering of extracellular pH suppresses low-Mg(2+)-induces seizures in combined entorhinal cortex-hippocampal slices.

- We are happy to replace the references as suggested by the Referee.

Referee #2

The authors have adequately addressed my comments, and I am satisfied with the revised version of the manuscript.